# Mechanisms of HIV-1 evasion to the antiviral activity of chemokine CXCL12 indicate potential links with pathogenesis

**Marie Armani-Tourret[1], Zhicheng Zhou[2], Romain Gasser[1], Isabelle Staropoli[2], Vincent Cantaloube-Ferrieu[1], Yann Benureau[2], Javier Garcia-Perez[3], Mayte Pérez-Olmeda[3], Valérie Lorin[4], Bénédicte Puissant-Lubrano[1], Lambert Assoumou[5], Constance Delaugerre[6], Jean-Daniel Lelièvre[7], Yves Lévy[7], Hugo Mouquet[4], Guillaume Martin-Blondel[1,8], Jose Alcami[3], Fernando Arenzana-Seisdedos[2¤], Jacques Izopet[1,9], Philippe Colin[1], Bernard Lagane[1] \***

**1** Infinity, Université Toulouse, CNRS, INSERM, UPS, Toulouse, France, **2** Viral Pathogenesis Unit, Department of Virology, INSERM U1108, Institut Pasteur, Paris, France, **3** AIDS Immunopathogenesis Unit, Instituto de Salud Carlos III, Madrid, Spain, **4** Laboratory of Humoral Immunology, Department of Immunology, INSERM U1222, Institut Pasteur, Paris, France, **5** INSERM, Sorbonne Université, Institut Pierre Louis d'Epidémiologie et de Santé Publique (IPLESP), Paris, France, **6** INSERM U944, Université de Paris, Hôpital Saint-Louis, APHP, Paris, France, **7** Vaccine Research Institute, INSERM and APHP, Hôpital H. Mondor, Créteil, France, **8** CHU de Toulouse, Service des Maladies Infectieuses et Tropicales, Toulouse, France, **9** CHU de Toulouse, Laboratoire de virologie, Toulouse, France

¤ Current address: Institut Pasteur of Shanghai, Chinese Academy of Sciences, Shanghai, China
* bernard.lagane@inserm.fr

**Data Availability Statement:** All relevant data are within the manuscript and its Supporting Information files, with the exception of the

## Abstract

HIV-1 infects CD4 T lymphocytes (CD4TL) through binding the chemokine receptors CCR5 or CXCR4. CXCR4-using viruses are considered more pathogenic, linked to accelerated depletion of CD4TL and progression to AIDS. However, counterexamples to this paradigm are common, suggesting heterogeneity in the virulence of CXCR4-using viruses. Here, we investigated the role of the CXCR4 chemokine CXCL12 as a driving force behind virus virulence. *In vitro*, CXCL12 prevents HIV-1 from binding CXCR4 and entering CD4TL, but its role in HIV-1 transmission and propagation remains speculative. Through analysis of thirty envelope glycoproteins (Envs) from patients at different stages of infection, mostly treatment-naïve, we first interrogated whether sensitivity of viruses to inhibition by CXCL12 varies over time in infection. Results show that Envs resistant (RES) to CXCL12 are frequent in patients experiencing low CD4TL levels, most often late in infection, only rarely at the time of primary infection. Sensitivity assays to soluble CD4 or broadly neutralizing antibodies further showed that RES Envs adopt a more closed conformation with distinct antigenicity, compared to CXCL12-sensitive (SENS) Envs. At the level of the host cell, our results suggest that resistance is not due to improved fusion or binding to CD4, but owes to viruses using particular CXCR4 molecules weakly accessible to CXCL12. We finally asked whether the low CD4TL levels in patients are related to increased pathogenicity of RES viruses. Resistance actually provides viruses with an enhanced capacity to enter naive CD4TL when surrounded by CXCL12, which mirrors their situation in lymphoid organs, and to deplete bystander activated effector memory cells. Therefore, RES viruses seem more likely to

nucleotide sequences of the CXCR4-using gp120s or gp160s derived from Envs #1, 6, 7, 16, 27, 28, 29, 32, 36, 41, 44 and 47, which have been deposited into GenBank (https://www.ncbi.nlm.nih. gov/genbank/) with the following accession numbers (MT951984; MT951985; MT951986; MT951987; MT951988; MT951989; MT951990; MT951991; MT951992; MT951993; MT951994; MT951995).

**Funding:** This work was supported by Agence Nationale de Recherches sur le SIDA et les hépatites virales (ANRS) (http://www.anrs.fr/fr) (Grants AAP-2013-2 and ECTZ63419 to B.L.), Institut National de la Santé et de la Recherche Médicale (INSERM) (https://www.inserm.fr), Institut Pasteur (https://www.pasteur.fr), Université Paul Sabatier Toulouse III (https://www.univ-tlse3. fr), The Milieu Intérieur Program (ANR-10-LABX-69-01 to H.M.) (http://www.milieuinterieur.fr) and Instituto de Salud Carlos III (projects RD16CIII/ 0002/0001 and PI19CIIII/0004 to J.A.) (https:// www.isciii.es). M.A.-T. was supported by a grant from Sidaction (https://www.sidaction.org/). The funders had no role in study design, data collection and analysis, decision to publish, or preparation of the manuscript.

**Competing interests:** The authors have declared that no competing interests exist.

deregulate CD4TL homeostasis. This work improves our understanding of the pathophysiology and the transmission of HIV-1 and suggests that RES viruses' receptors could represent new therapeutic targets to help prevent CD4TL depletion in HIV+ patients on cART.

## Author summary

HIV-1 infects immune cells by binding CD4 and a coreceptor, CCR5 or CXCR4. CXCR4-using viruses are thought to accelerate depletion of CD4 T lymphocytes (CD4TL) and AIDS development. This paradigm, while often true, is however not seen in all patients, suggesting heterogeneity in the virulence of viruses. Here, we show that CXCR4-using viruses can be discriminated on the basis of their resistance to the anti-HIV-1 effect of the CXCR4 chemokine CXCL12. Resistant (RES) viruses are found in patients with low CD4TL levels. They are featured by unique properties of their envelope glycoprotein and the way they use CXCR4. Our data also indicate that RES viruses could contribute to CD4TL depletion. They more effectively kill bystander activated effector memory CD4TL, a subset of CD4TL enriched in HIV+ patients, and enter CD4TL subsets surrounded by CXCL12. Resistance could therefore enhance the ability of viruses to target naive cells and their precursors in lymphoid organs where CXCL12 is present, further hindering CD4TL renewal. Beyond presenting a novel view of CXCR4-using HIV-1 pathogenesis, this work also opens new therapeutic perspectives and increases our knowledge of the still debated role of CXCL12 in HIV-1 transmission.

## Introduction

The natural course of HIV-1 infection results in the destruction of CD4+ T lymphocytes (CD4TL), the onset of AIDS and death from opportunistic infections. HIV-1 entry into CD4TL relies on binding of the gp120 subunit of its envelope glycoprotein (Env) to CD4 and a chemokine receptor used as coreceptor, CCR5 or CXCR4. The gp41 subunit of Env then inserts into the membrane of the CD4TL and triggers fusion between the virus and the cell [1]. CCR5-using viruses (R5-tropic) are more frequently transmitted [2] and are predominant in patients. At the time of primary HIV-1 infection (PHI), the viruses that can use CXCR4, exclusively (X4-tropic) or in addition to CCR5 (R5X4- or dual-tropic), are rare (< 10%) [3], but their proportion increases as infection progresses [4–6]. The pathogenesis of infection has often been linked to HIV tropism. CXCR4-using viruses are more often than R5-viruses associated with accelerated depletion of CD4TL and progression to AIDS [7,8]. The increased pathogenicity of CXCR4-using viruses is explained by their broader cellular tropism, in particular for naive CD4TL that express CXCR4 but not CCR5 [9–11]. In monkey models of infection, CXCR4 usage is associated with rapid depletion of naive cells in the blood and secondary lymphoid organs (SLOs) [12,13]. Infection with CCR5-using viruses is confined to memory cells, and may eventually lead to depletion of naive cells, but more progressively [14]. In HIV infection, loss of quiescent cells in SLOs has been linked to depletion of circulating CD4TL and incomplete immune reconstitution upon antiretroviral therapy [15,16]. Most of these cells express CXCR4 but only moderately CCR5, explaining the increased pathogenicity of CXCR4-using viruses compared to R5 HIV-1. CXCR4-using viruses can also interfere with the replenishment of naive cells through depletion of thymocytes [17,18] and infection of CD34 + multipotent hematopoietic progenitor cells in the bone marrow (BM) [19].

There are, however, many counter-examples to the paradigm of increased pathogenicity of CXCR4-using viruses. Although they are more common in patients with low CD4TL count, they are also present in immunocompetent patients [5]. Similarly, the presence of CXCR4-using viruses in the very early stages of infection does not necessarily translate into accelerated progression of the disease [20,21]. These data therefore indicate that there is heterogeneity in the virulence of CXCR4-using viruses, the underlying mechanisms of which, however, remain unknown. To get better insights into this issue, we hypothesized that the CXCR4 chemokine CXCL12 might act as a driving force in evolution of the phenotypic properties of CXCR4-using viruses. CXCL12 binding to CXCR4 activates a variety of cellular functions and regulates trafficking and homeostasis of leukocytes [22]. *In vitro*, CXCL12 inhibits viral entry by preventing Env binding to CXCR4 and triggering endocytosis of the receptor [23], suggesting that it could also act as a constraint for CXCR4-using viruses in patients. We speculated that the viruses could evolve to overcome this constraint during infection, and that this could influence their properties including pathogenicity.

To test our hypothesis, we investigated whether the sensitivity of viruses to inhibition by CXCL12 changes during the course of infection. We studied this in infection assays of primary CD4TL and/or T-cell lines with a panel of viruses pseudotyped with X4 or R5/X4 Envs from the blood of patients, mostly treatment naive, isolated at different stages of infection (PHI, chronic or AIDS). Results indicate that CXCL12 resistant (RES) viruses are common in patients with a severe drop in peripheral blood CD4TL levels, most often late in infection, in one case at the time of PHI. Compared to CXCL12 sensitive (SENS) viruses, which account for almost all chronic-stage viruses, RES viruses have a greater potential to deregulate homeostasis of CD4TL, which may explain the striking association between their presence in patients and increased CD4TL depletion. These viruses also have unique characteristics, which could be exploited to eliminate them selectively. They differ from SENS viruses by the conformation/antigenicity of their Env and the fact that they use subsets of CXCR4 molecules with little or no access to CXCL12. Targeting these receptors may offer new opportunities to tackle viruses at risk for the patient while preserving the homeostatic functions of the CXCL12/CXCR4 pair. While these data define CXCL12 as a major player in the pathogenesis of CXCR4-using viruses, they however raise questions about its role in HIV-1 transmission.

## Results

### Viruses with increased resistance to CXCL12 are common in patients with low CD4 T-cell count

We tested the capacity of CXCL12 to inhibit infection of CD4TL with viruses pseudotyped with Envs of virus clones isolated from PBMCs of 5 patients (P) of the Amsterdam cohort (ACS) (**Figs 1A and S1A and S1B**), three treatment naïve, two under AZT monotherapy (**S1 Table**). For each patient, PBMCs were collected early after the appearance of CXCR4-using viruses (mean CD4TL count = 438/μl) and at the time of AIDS (late) (62 CD4TL/μl). CXCL12 potently inhibited the early viruses but one (P704#44) and the control strain NL4-3, as revealed by low $IC_{50}$s for the chemokine, within the range of its reported affinity constant for CXCR4 on CD4TL (from a few to tens of nM [24], blue dotted lines in **Fig 1A**). However, it inhibited less potently the late viruses, suggesting that X4 viruses evolve resistance to CXCL12 in the course of infection. By contrast, viruses pseudotyped with early (566 CD4TL/μl) or late (108 CD4TL/μl) R5 Envs from 5 other treatment-naïve patients of the ACS (**S2 Table**) did not change in their sensitivity to the CCR5 chemokines CCL4 (**Fig 1B**), CCL3 and CCL5 (**S1C and S1D Fig**). It should be noted that previous work has already reported a statistically significant decrease in the sensitivity of late R5 viruses to CCL5, not necessarily to CCL3 or 4 [25]. In

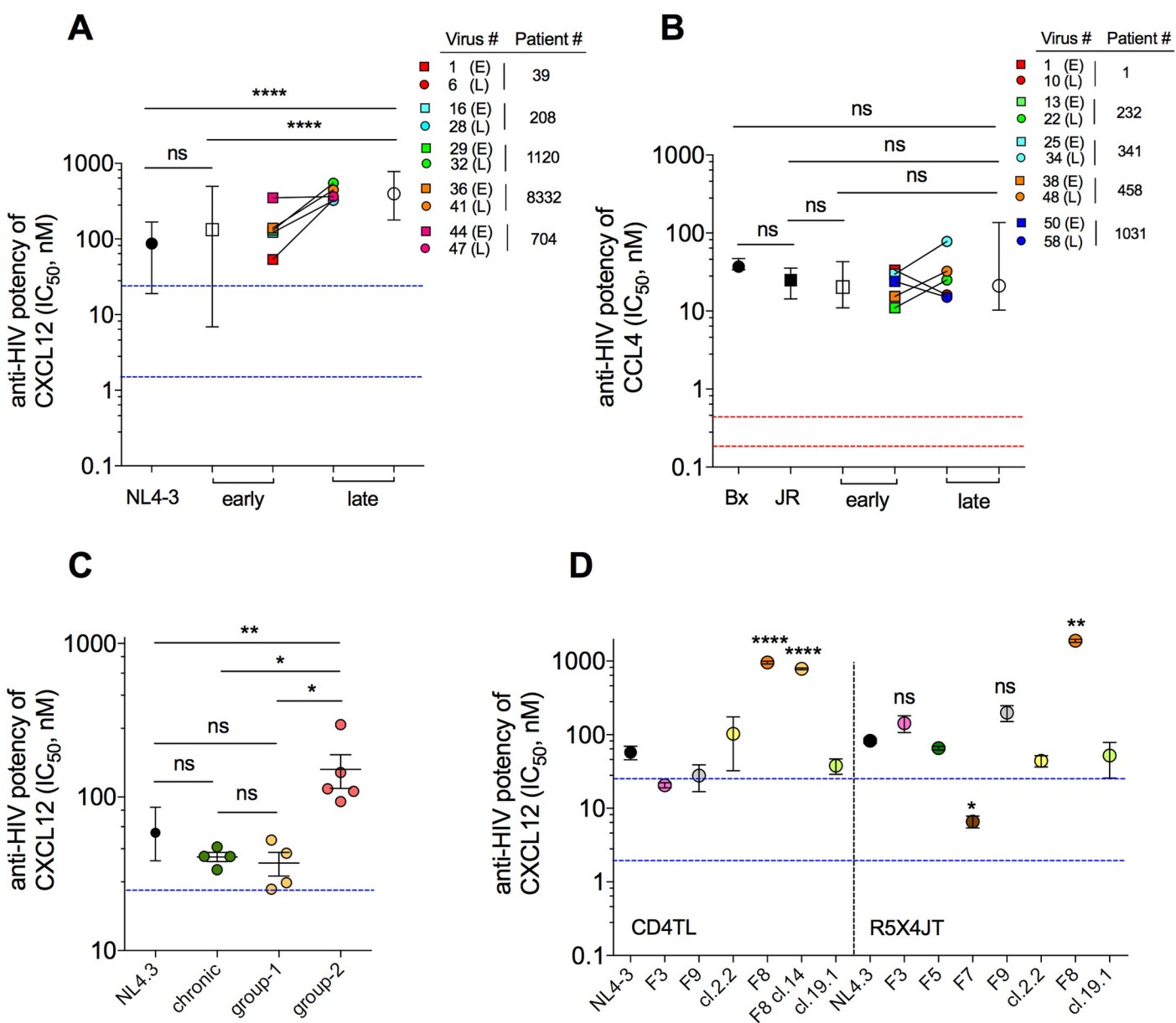

**Fig 1. CXCL12 resistant viruses are frequent in patients with low CD4 T-cell count. A** Viruses pseudotyped with early (E) or late (L) X4 Envs from 5 patients of the ACS were used to infect CD4TL w/or w/o CXCL12 at increasing concentrations. Mean (± range) (open symbols) and individual (colored symbols) $IC_{50}$s of CXCL12 are shown. The same color code as shown here is used throughout the article. Blue dotted lines represent the range of CXCL12/CXCR4 affinities. The X4 HIV-1 strain NL4-3 was used as control. **B** Mean and individual $IC_{50}$s for neutralization by CCL4 of viruses pseudotyped with early (E) or late (L) R5 Envs from 5 patients of the ACS. Red dotted lines represent the range of CCL4/CCR5 affinities. The JR-FL and Bx08 strains are controls. **C** CXCL12-mediated inhibition of recombinant viruses pseudotyped with Envs from plasma of patients diagnosed at a chronic (green symbols) or late stage (yellow and red symbols) of infection. Group-2 and group-1 comprise Envs that are, or are not, more resistant to CXCL12 than NL4-3 or chronic Envs, respectively. Each data point represents the mean $IC_{50}$ of CXCL12 for neutralization of a given virus. **D** CXCL12-mediated neutralization of recombinant viruses expressing Envs isolated from plasma of patients at the time of PHI. Each data point is the mean $IC_{50}$ of CXCL ($\pm$ SEM) for the indicated virus, calculated from independent experiments on CD4TL (left) and/or R5X4JT cells (right). Statistics are relative to NL4-3.

common with the previous data however, we show here that the CCR5 chemokines have weak antiviral potencies, as judged by $IC_{50}$s that are much higher (up to $\geq$ 100-fold) than their binding affinities for CCR5 (red dotted lines in **Fig 1B**). They also weakly inhibited R5 viruses from PHI (**S1C and S1D Fig**). These results considered altogether with our previous works [26,27]

indicate that R5 viruses share the capacity to escape inhibition by CCR5 chemokines, regardless of the stage of infection.

To confirm that CXCL12-resistant (RES) HIV variants are present at an advanced stage of infection, we next measured the effects of CXCL12 on CXCR4-using, recombinant viral populations generated from plasma of nine late diagnosed individuals with CD4TL count < 200/µl (**Fig 1C** and **S3 Table**). Prior tropism testing on CCR5- and CXCR4-expressing Jurkat T cells (R5X4JT cells) indicated that three viral populations contained pure X4 viruses (samples EBTY, ZEYB and IFQA), the others comprising Dual/Mixed- (D/M-) tropic viruses (**S1E Fig**). We performed the inhibition assays on the same cells, because they led to higher infectivity levels than primary CD4TL. Experiments were run in the presence of the CCR5 antagonist maraviroc (MVC, 10 µM) to force R5X4 viruses use CXCR4 only. Compared to NL4-3, half of the viruses were statistically more resistant to inhibition by CXCL12 (group-2 in **Fig 1C**), while the remaining ones were either equally or more potently inhibited (group-1). Unrelated recombinant viral populations from plasmas of patients at the chronic stage of infection were also highly sensitive to CXCL12 (**Fig 1C**). Overall, these results confirm that viruses with increased resistance to CXCL12 are frequent at a late stage of infection in patients with low CD4TL levels. Some patients (*i.e.* patients in group-1), however, show CD4TL depletion in the absence of RES viruses (**S1F Fig**).

We next investigated whether RES viruses are also present at the time of PHI. As CXCL12 was proposed to counter-select transmission of CXCR4-using viruses [28,29], we speculated that the rare cases of these viruses that are transmitted could be resistant to the chemokine. Five recombinant viral populations (F3, F5, F7, F8 and F9) and three viral clones (2.2, F8 cl.14 and 19.1) obtained from plasmas of seven patients with PHI (**S4 Table**) were tested for their sensitivity to CXCL12 on CD4TL and/or R5X4JT cells (**Fig 1D**). As before, experiments were performed in the presence of MVC, because with the exception of 19.1, which is purely X4, all other viral samples are D/M-tropic (**S1G Fig**). Surprisingly, only population F8 and the clone derived from it (F8 cl.14) were resistant to CXCL12, the others viruses being as sensitive to CXCL12 as was NL4-3 (**Figs 1D** and **S1H**). Medical monitoring indicated that Patient F8, in contrast to the other patients, suffered from a severe depletion in blood CD4TL at the time of diagnosis (CD4TL count = 45/µl), which was only partly corrected after 12 months of effective therapy (see paragraph on Patient characteristics). Patient F8 thus reproduces the situation of patients late in infection, where RES viruses are present in a context of low CD4TL count. This suggests that there could be a link between the presence of RES viruses and increased depletion of CD4TL. In the following, we studied the mechanisms of resistance to CXCL12 and how it may be related to increased pathogenesis of the infection.

## Resistance to CXCL12 maps to gp120, not to gp41

Experiments showed that differences in Env incorporation into the pseudotyped viruses do not explain the differences in their sensitivity to inhibition by CXCL12 (**S2 Fig**). We therefore speculated that resistance to CXCL12 could owe to increased efficacy of any of the viral entry steps. Our data suggest that changes in the fusion process are not involved. Substituting the gp41 subunit in RES Envs by gp41 of the CXCL12-sensitive strain NL4-3 did not influence resistance (**S1I-S1K Fig**). We thus investigated whether RES viruses have improved interactions with CD4 and/or CXCR4.

## Resistance to CXCL12 is not due to improved binding to CD4

We first determined the affinity constants Ki of gp120s for CD4 (**Fig 2A**). We measured the capacity of gp120s to compete with the mAb Q4120 for binding to CD4 [30]. Late X4 gp120s,

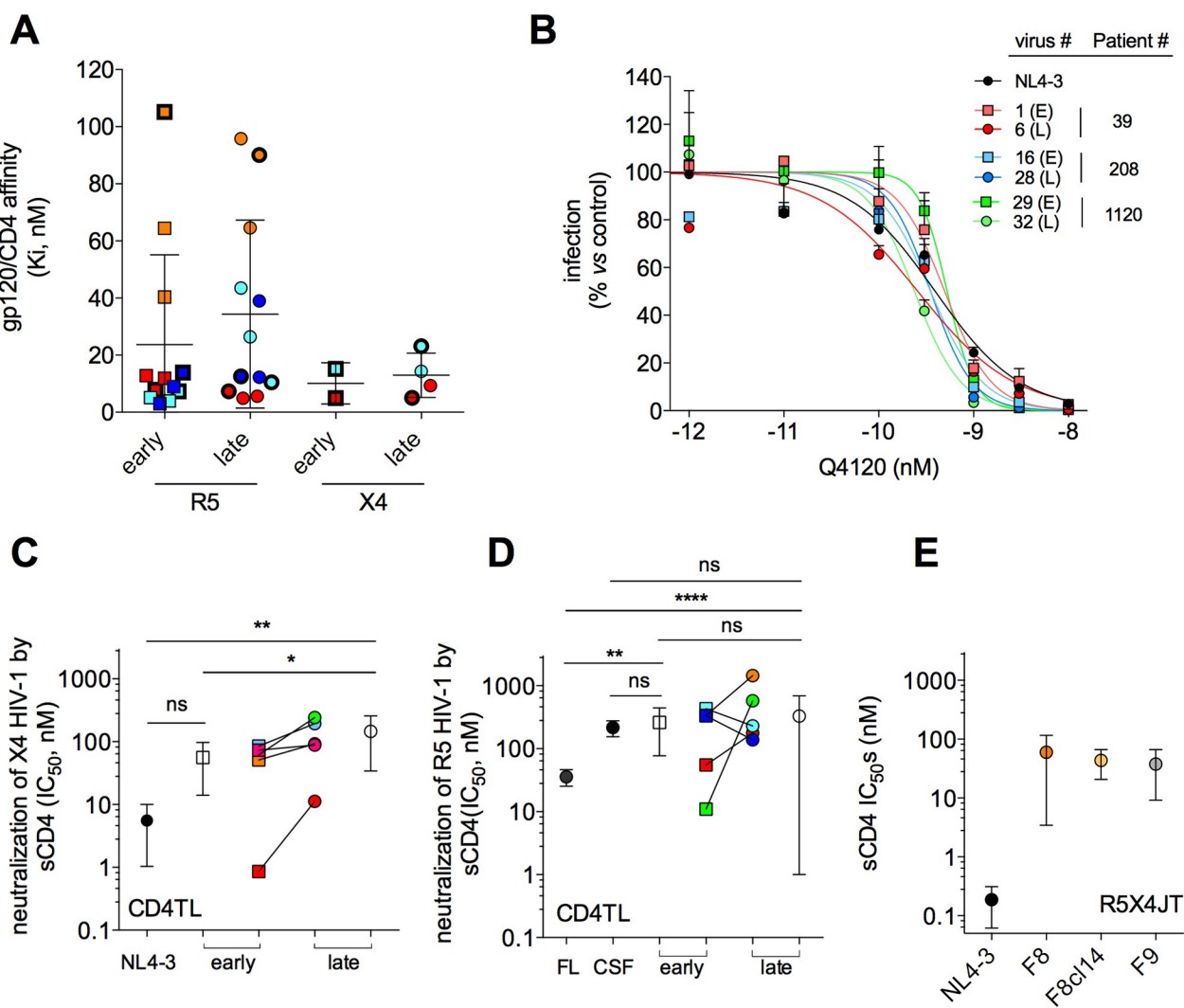

**Fig 2. Resistance to CXCL12 is not due to improved Env/CD4 interactions. A** Equilibrium dissociation constants Ki of interactions between CD4 and early or late R5 or X4 gp120s. Each data point is the mean Ki value for a given gp120. Color code: R5 gp120s: Red, P#1 (Early (E): # **1**, 2, 3; Late (L): # **10**, 11, 12); Light blue, P#341 (E: # **25**, 26, 27; L: # **34**, 35, 36); Orange, P#458 (E: # 37, **38**, 39; L: 46, 47, **48**); Dark blue, P#1031 (E: # 49, **50**, 51; L: # **58**, 59, 60). X4 gp120s: Red, P#39 (E: # **1**; L: # **6**, 7); Light blue, P#208 (E: # **16**; L: # 27, 28). Symbols in bold refer to gp120s from the Envs shown in **Fig 1A and 1B**. Error bars are SEM. **B** Inhibition by mAb Q4120 of infection of CD4TL with viruses pseudotyped with early (E) or late (L) Envs. Results are expressed as percent infection relative to control infection in the absence of Q4120. One representative experiment is shown (error bars: SEM). **C** IC50s for neutralization by sCD4 of infection of CD4TL with viruses pseudotyped with the early or late X4 Envs shown in **Fig 1A**. Mean (± SD) (open symbols) and individual (colored symbols) IC50s are shown. **D** Neutralization by sCD4 of the R5 Envs shown in **Fig 1B**. JR-FL and JR-CSF Envs were included as controls. **E** Neutralization by sCD4 of F8, F8 cl. 14, F9 and NL4-3 Envs. Results are means ± SD of experiments on R5X4JT cells.

including gp120s from the RES viruses shown in **Fig 1A**, did not differ from their early counterparts in the capacity to bind CD4. Both types of X4 gp120s bind CD4 with high affinity, as revealed by Ki values of few nM. In the same line, viruses pseudotyped with early and late X4 Envs showed similar sensitivity to inhibition by Q4120 (**Fig 2B**). Affinities of X4 gp120s for CD4 are also similar to those of R5 gp120s (**Fig 2A**). Overall, these results indicate that affinity of gp120/CD4 interactions does not vary with viral tropism and the stage of infection.

The capacity of HIV-1 to engage CD4 also depends on equilibrium between different conformational states of the Env trimer. Envs that are more likely to adopt an open conformation (e.g. JR-FL Env) have improved interactions with CD4 and are more sensitive to inhibition by

soluble CD4 (sCD4) [31]. We therefore reasoned that if resistance to CXCL12 stems from improved engagement of CD4, then RES viruses should be more sensitive to inhibition by sCD4. However, the reverse situation was observed (**Fig 2C**). Late X4 Envs were three-fold less sensitive to inhibition by sCD4 than early Envs ($IC_{50}$ = 56 *vs* 146 nM, *P* = 0.016). In contrast, early and late R5 Envs exhibited similar sensitivity to sCD4, in the range of that of the JR-CSF strain (**Fig 2D**), which is notoriously resistant to sCD4 [32]. Of note, the X4 Envs from patient #39 (red symbols in **Fig 2C**) were unusually highly sensitive to sCD4. However, the reasons for the intra- and inter-individual diversities in sensitivity of X4 Envs to sCD4 are unclear. In particular, all Envs comprise Leu193 and Ile423 that stabilize the Env trimer in a closed, sCD4-resistant, conformation [31,33]. Finally, the CXCR4-using Envs isolated during PHI also showed lower sensitivity to sCD4 (**Fig 2E**), a property that is actually common to transmitted viruses [34]. Collectively, these data suggest that resistance to CXCL12 occurs in a context where the Env trimer adopts a more closed conformation.

## Env antigenicity varies between SENS and RES viruses

The above results suggested that SENS and RES Env trimers experience distinct conformational dynamics. To explore this, we tested their neutralization sensitivity to broadly neutralizing, human monoclonal antibodies (bNAbs) targeting quaternary epitopes of the Env trimer (**Fig 3**) [35]. We speculated that divergent conformational constraints between RES and SENS Envs could translate into differential sensitivities to bNAbs. In neutralization assays on TZM-bl cells, three out of the five RES Envs of the ACS were insensitive to PGT145 (**Fig 3A**), which recognizes the V1/V2 region at the trimer apex, as well as to the related bNAbs PG16 and PGDM1400 (**S3A and S3B Fig**). The three Envs lack the N160 glycosylation site in the V2 loop (**S3C Fig**), which is the main contact site for V2-apex bNAbs [36]. RES Env from P#208 was also more resistant to these bNAbs, compared to its SENS counterpart. While it has retained N160, it lacks R166 that is also important for binding PGT145-type bNAbs [37]. The last RES Env of the ACS (from P#704) also tended to be more resistant to PG16 and PGDM1400, but not to PGT145, compared to the SENS Env from the same patient. This Env shows a N-to-K substitution at position 156 (**S3C Fig**). This is consistent with previous work showing that the N156 glycan may play a role in neutralization activity of V2-apex bNAbs [38].

The RES Envs of the ACS therefore have in common that they are more resistant to V2-apex bNAbs, compared to SENS Envs. F8 Env, in contrast to these other RES Envs, did not show improved resistance to the V2-apex bNAbs (**Figs 3A and S3A and S3B**), although it hosts several mutations of residues that have been reported to regulate their binding (N156K, R166G, K171E) (**S3C Fig**) [37,39]. However, F8 was insensitive to PGT151 (**Fig 3B**), which recognizes a complex epitope defined by inter- and intra-protomer interactions involving both gp120 and gp41 at the base of the trimer [40]. Several features in the sequence of F8 Env could explain this result, such as the lack of the N295 glycan at the base of V3, a K-to-E substitution at position 490 in the C5 region of gp120, and several mutations within the HR2 region of gp41 (S640H and L and N residues at positions 641 and 644) (**S3D Fig**) [37,40]. Finally, SENS and RES Envs did not differ in their sensitivity to bNAbs against the CD4 binding site (VRC01), the glycan-V3 region (PGT121), or the MPER domain (10E8) (**Fig 3C-E**).

In summary, we show that RES Envs have in common alterations in the epitope of V2-apex bNAbs that may or may not translate into resistance to neutralization. In the case of F8, these alterations are accompanied by changes in the gp120/gp41 interface. Importantly, both regions critically regulate equilibrium between distinct conformations of the Env trimer [41,42]. Our results therefore support our hypothesis that SENS and RES Env trimers display distinct conformational dynamics.

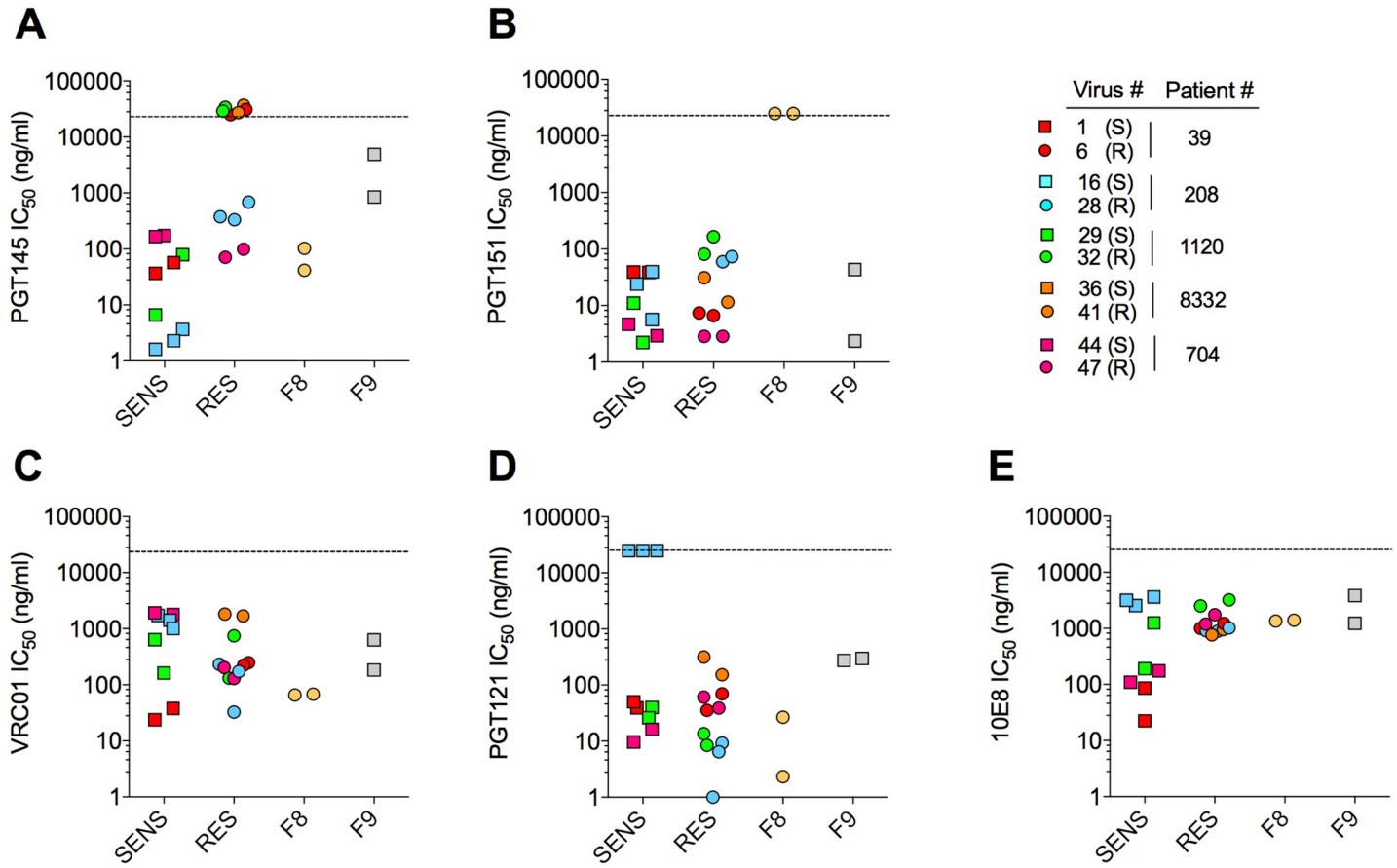

**Fig 3. Sensitivity of CXCL12-SENS and CXCL12-RES viruses to broadly neutralizing antibodies.** Viruses pseudotyped with SENS, RES, F8 or F9 Envs were exposed to bNAbs and then incubated with TZM-bl cells. For each Env/bNAb pair, the $IC_{50}$s from two or three independent experiments are shown. The dashed line indicates the highest antibody concentration used (25,000 ng/ml). A data point on the dashed line indicates that $IC_{50} > 25,000$ ng/ml.

## CXCL12 is impaired in its ability to block the binding of RES viruses to CXCR4

Both steric blockade of virus binding to CXCR4 and endocytosis of the coreceptor contribute to the antiviral effects of CXCL12. We thus investigated which of these two mechanisms is defective in the context of resistance to CXCL12. To this end, we repeated infection assays of CD4TL with RES or SENS viruses in the presence of P2G, a CXCL12 variant that binds CXCR4 with comparable affinity compared to CXCL12 but does not internalize the receptor [43–45]. As previously shown [43], P2G had a reduced anti-HIV-1 potency compared to CXCL12 (**S4 Fig**), presumably due to lack of CXCR4 endocytosis [23]. However, similarly to CXCL12, P2G inhibited RES viruses less potently than SENS viruses. The magnitude to which the antiviral potencies are reduced is comparable for both chemokines (**S4 Fig**), suggesting that resistance to CXCL12 originates from its reduced capacity to competitively inhibit virus binding to CXCR4.

## RES viruses use CXCR4 receptors with low affinity for CXCL12

We therefore investigated whether RES Envs have increased binding affinity for CXCR4, compared to SENS Envs. Previous experiments on purified CXCR4 indicated low affinity

(equilibrium dissociation constant Kd > 100 nM) of lab-adapted Envs for the coreceptor [46]. But, affinity of primary Envs for CXCR4 in its native environment is still not known. Here, we measured saturation binding of $^{35}$S-labeled gp120/sCD4 complexes to membrane preparations from unstimulated PBMCs (**Fig 4A**). CXCL12-SENS and RES gp120s bound CXCR4 with apparently similar low affinities (Kd = 115 and 123 nM, respectively). Remarkably, SENS gp120s consistently bound more CXCR4 molecules than RES gp120s (**Fig 4A**). Similarly, we recently reported that distinct R5 gp120s can recognize different amounts of CCR5 [32], because they bind differentially to distinct conformations of the receptor, which exist in different quantities at the cell surface [32]. Monoclonal antibodies (mAbs) also showed divergent binding levels to CCR5 [47] or CXCR4 [48,49]. Our data in light of these previous results thus suggest that although CXCL12-RES and SENS gp120s bind CXCR4 with similar affinity, they probably do not bind the same CXCR4 molecules.

We thus hypothesized that resistance to CXCL12 might be related to binding of Env to CXCR4 molecules with low affinity for the chemokine. We performed competition binding assays of SENS or RES $^{35}$S-gp120s by CXCL12 (**Fig 4B and 4C**). The concentrations of both gp120s were chosen so that they have similar binding levels to CXCR4 in the absence of CXCL12 (inset of **Fig 4B**). Under these conditions, CXCL12 displaced RES gp120s less potently and with lower efficacy than SENS gp120s. Higher concentrations of CXCL12 were required to displace the binding of RES gp120s compared to SENS gp120s, suggesting that RES gp120s interact with CXCR4 molecules that have a lower affinity for CXCL12 than those interacting with SENS gp120s. Furthermore, RES gp120s were only partially inhibited at the highest CXCL12 concentration tested (1 μM), suggesting that they also interact with a fraction of CXCR4 that does not bind CXCL12 at all.

We next characterized the nature of the CXCR4 molecules that bind both types of gp120s. We previously reported that CCR5 when uncoupled from G proteins has low affinity for chemokines but retains high affinity for R5 gp120s [26], a finding that we extend here to a wider variety of gp120s (**S5A Fig**). However, uncoupling CXCR4 from G-proteins only marginally influences the binding of CXCL12 and X4 gp120s (**S5B Fig**). This indicates that the low affinity receptors for CXCL12 does not stem from uncoupling from G proteins.

We then investigated whether CXCR4 molecules with varying affinity for CXCL12 represent distinct antigenic conformations of the receptor [48,49]. We found that distinct antigenic forms of CXCR4 exist on CD4TL, as revealed by divergent binding levels of distinct anti-CXCR4 mAbs to these cells (**S5C Fig**). These mAbs bound CXCR4 with high affinity (Kd of few nM or lower) but some of them (44717, 44716 and 44708) labeled more CXCR4 molecules than others (4G10 or 12G5). These mAbs were shown to target distinct regions of CXCR4, *i.e.* the N-terminus (4G10 [49,50]), extracellular loop 2 (ECL2) (MAbs 44717, 44716 and 44708 [49]) or a conformational epitope comprising residues of the N-terminus, ECL2 and ECL3 (12G5 [49]). The fact that these mAbs recognize different amounts of CXCR4 molecules on CD4TL suggests that the receptor adopts multiple conformations that differ in the nature of the epitopes they expose [48]. We then explored the capacity of these mAbs to inhibit infection of CD4TL with SENS or RES-viruses (**Fig 4D**). Even when used at saturating concentration (10 μg/ml), some (12G5, 716 and 708) failed to inhibit infection. This was unexpected, because these mAbs target regions of CXCR4 that also bind HIV-1 Env (in particular the N-terminus and/or ECL2) [49]. Again, this suggests that the epitopes of these mAbs are not exposed on the CXCR4 conformations, which are used by viruses. Similarly, we [32] and others [47] reported that R5 viruses use only subsets of the CCR5 molecules for entry into cells. Interestingly, mAb 717 inhibited infection by SENS, but not RES, viruses (**Fig 4D**), confirming that both viruses do not use the same CXCR4 conformations. Finally, we observed that mAb 4G10 efficiently inhibits infectivity of SENS and RES viruses, albeit partially (> 60%), suggesting that a fraction

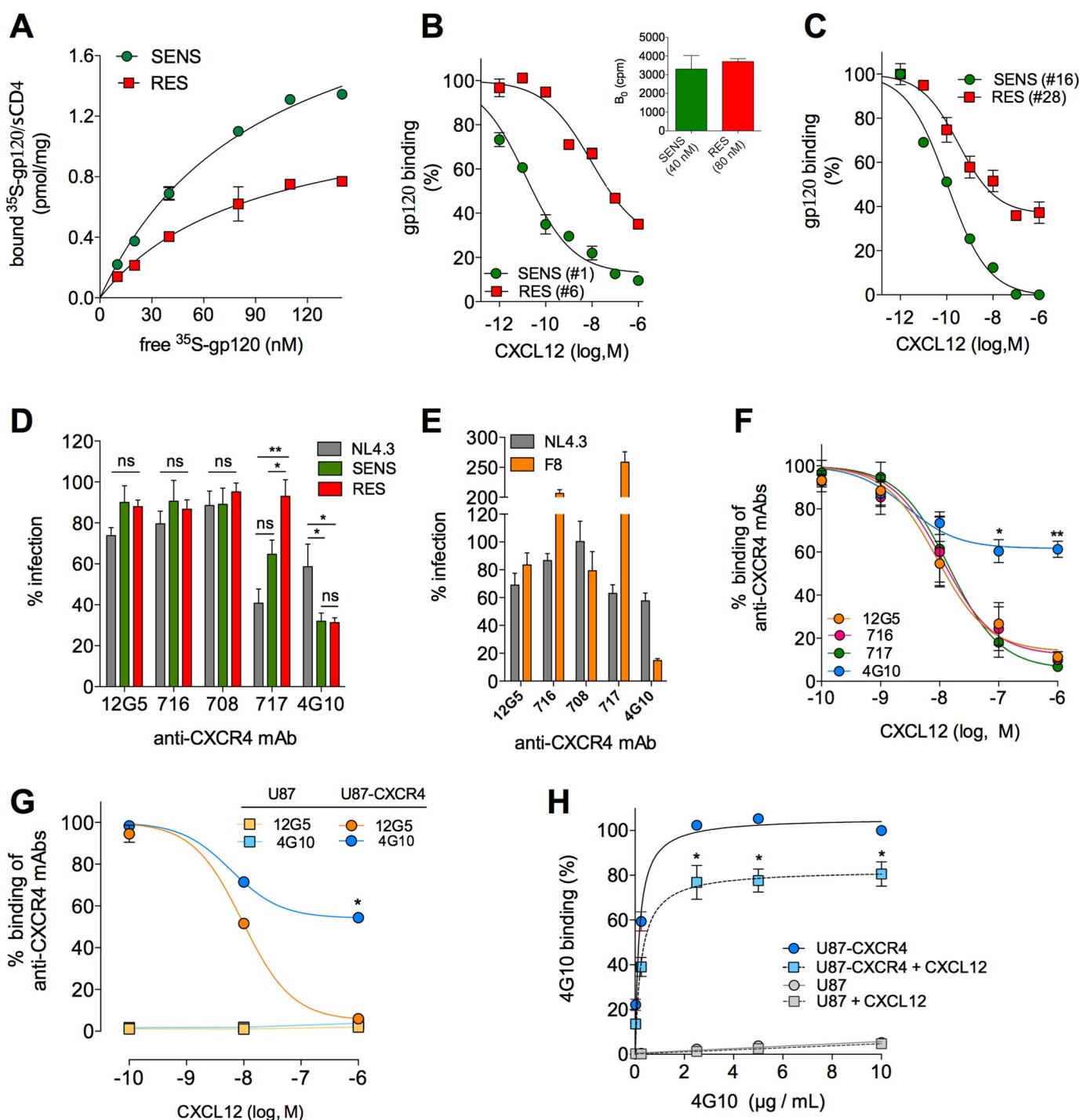

**Fig 4. CXCL12-resistant viruses interact with CXCR4 molecules that are weakly accessible to the chemokine. A** Specific binding (SB) of gp120/sCD4 complexes to PBMC membranes. A representative experiment with gp120s #1 (SENS) and #6 (RES) (P#39) is shown. Gp120 #16 (SENS) and #28 (RES) (P#208) gave similar results. The extrapolated $B_{max}$ (i.e. maximum level of binding) were 2.9 and 1.6 pmole/mg for SENS and RES gp120s, respectively. **B** and **C** Displacement of gp120 binding by CXCL12. Forty and 80 nM of SENS or RES gp120/sCD4, respectively, were used, leading to similar levels of SB in the absence of CXCL12 (inset). **D** Effects of anti-CXCR4 mAbs on infection of CD4TL with SENS and RES viruses, expressed as percent infection relative to untreated control (100%). Means ± SEM of 5 independent experiments are shown, carried out with 20 ng of p24 of the viruses pseudotyped with Envs # 1 (SENS), # 6 (RES), # 16 (SENS), # 28 (RES) or NL4-3 Env. **E** Effects of anti-CXCR4 mAbs on virus population F8. Ten μM MVC was added to the assay medium. **F** Displacement of binding of anti-CXCR4 mAbs to CD4TL by CXCL12. Results (means ± SEM) are expressed relative to binding in the absence of CXCL12 (100%). The competition curves gave Ki values for CXCL12 ranging between 2.6 and 8.7 nM. **G** Displacement of binding of mAbs 4G10 and 12G5 to U87-CXCR4 cells by CXCL12. **H** Specific binding of 4G10 to U87-CXCR4 cells in the presence or absence of 1 μM CXCL12. Non-specific binding on parental U87 cells is also shown (grey

symbols). Results are expressed as percent of maximal binding of 4G10 in the absence of CXCL12. 4G10 had similar affinity constants in the presence or absence of CXCL12 (Kd = 0.16 *vs* 0.25 μg/ml). Statistics are relative to binding w/o CXCL12.

of receptors binding the viruses are not recognized by the mAb. The mAb also inhibited the CXCL12-resistant virus population F8 ($>$ 80%) (**Fig 4E**), while for unknown reasons mAbs 716 and 717 enhanced its infectivity. Similar paradoxical effects have already been reported [49].

The capacity of mAb 4G10 to inhibit SENS and RES viruses suggested that to some extent it could recognize both high- and low-CXCL12 affinity conformations of CXCR4. We thus measured the ability of CXCL12 to compete with 4G10 and other anti-CXCR4 mAbs for binding to CD4TL (**Fig 4F**) or CXCR4-expressing U87 cells (**Fig 4G**). CXCL12 fully inhibited MAbs 12G5, 716 and 717 with affinity constants (Ki) in the nM range (legend of **Fig 4F**). CXCL12 also potently displaced the binding of mAb 4G10, but only partially. Remaining binding of 4G10 in the presence of excess CXCL12 represented binding to CXCR4, because the mAb did not bind to cells lacking the receptor (**Fig 4G**). Saturation experiments further showed that 1 μM CXCL12 decreased the maximal level of binding of 4G10 with no change in its affinity for CXCR4 (**Fig 4H**). This suggests that 4G10 binds with the same affinity to at least two types of CXCR4 molecules, one with high affinity for CXCL12, the other not accessible to the chemokine. Binding of 4G10 to high- and low-CXCL12 affinity receptors may explain why the mAb inhibits as efficiently SENS and RES viruses (**Fig 4D**).

## The degree of resistance to CXCL12 distinguishes highly pathogenic from less pathogenic viruses

We next investigated whether viral resistance to CXCL12 could be responsible for increased depletion of CD4TL in patients. We first evaluated whether resistance may have altered the tropism of viruses for T-cell subsets. We compared the capacity of SENS and RES viruses to fuse with naive (Tn), central memory (Tcm), effector memory (Tem) and terminally differentiated Tem (Temra) CD4TL (**Fig 5A**). PHA/IL-2-treated CD4TL were inoculated with β-lactamase (BLaM)-vpr-containing viruses, loaded with the fluorescent BLaM substrate CCF2 and labeled with the anti-CCR7 and anti-CD45RA mAbs. The percentage of cells that have fused with the viruses (cells with cleaved CCF2) was then determined by flow cytometry (**S6A Fig**) and among these cells, the proportion of Tn, Tcm, Tem and Temra cells was compared to their proportion among all cells (**S6B Fig**). The calculated proportion ratios (PR) for each cell subset did not differ between SENS viruses, RES viruses and NL4-3, indicating that the viruses have similar cellular tropisms (**Fig 5A**). They all showed a tropism shift toward Tcm cells (PR value $>$ 1) at the expense of Tn and Temra cells (PR $<$1), while the proportion of Tem cells was unchanged among fused cells, as compared to all cells (PR = 1). As control, fusion of the R5 strain JR-FL increased from Tn to Temra cells, in line with the increase in CCR5 expression level (**S6C Fig**) and proportion of CCR5 positive cells (**S6D Fig**). In contrast, expression of CXCR4 and CD4 was similar among T-cell subsets.

We next repeated the fusion experiments in the presence of CXCL12 (**Fig 5B**). SENS viruses were similarly inhibited regardless of the nature of CD4TL subsets (65–70% inhibition at 300 nM CXCL12). RES viruses were much less inhibited by the chemokine, in particular when fusing with Tn (and to a lesser degree with Tcm) cells. Actually, the fusion of RES viruses with Tn cells was virtually unaffected by CXCL12. These results indicate that CXCL12 effectively protects the CD4TL subsets from fusion with SENS viruses, but does so less efficiency or even not at all with RES viruses.

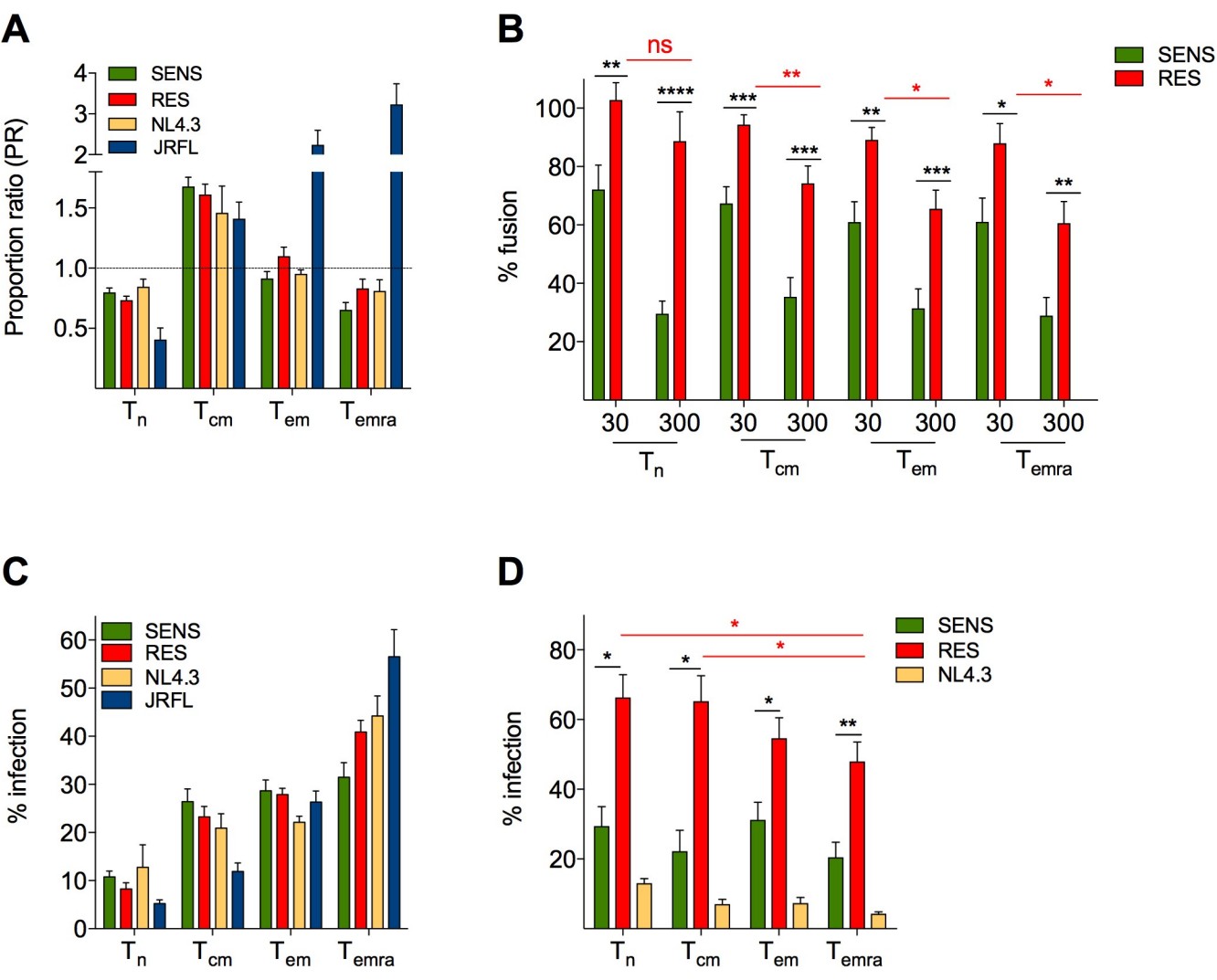

**Fig 5. CXCL12 protects CD4TL subsets against SENS viruses, but not against RES viruses. A** Tropism of SENS and RES viruses for Tn, Tcm, Tem and Temra CD4TL. For each cell subset, the panel represents its proportion (PR) within CD4TL that have fused with viruses, compared to its proportion among all cells (infected and uninfected cells). BLaM-vpr-containing viruses pseudotyped with the SENS or RES Envs of P#39, #208, #1120, #8332 and #704 were used. NL4.3 (X4) and JRFL (R5) Envs served as controls. Results are means ± SEM of three independent experiments. **B** Fusion of SENS and RES viruses with CD4TL subsets in the absence or presence of 30 or 300 nM CXCL12. Percent fusion for each virus is expressed relative to fusion in the absence of CXCL12 (100%). **C** Infectivity of SENS and RES viruses in sorted CD4TL subsets. Percent infection of each T cell subset is expressed relative to infection of all CD4TL subsets (100%). Means ± SEM of three experiments are shown. **D** Infection experiments were repeated in the presence or absence of 300 nM CXCL12. Results are expressed relative to infectivity in the absence of CXCL12 (100%).

We also FACS-sorted the different CD4TL subsets with anti-CCR7 and CD45RA mAbs and then subjected them to single-cycle infection with SENS viruses, RES viruses, JR-FL or NL4-3 (**Fig 5C**). Viral infectivity increased from Tn to Temra cells, most likely due to the increase in the proportion of activated cells [51], but did not change between SENS and RES viruses, confirming their common cellular tropism. In the presence of CXCL12 (**Fig 5D**), SENS viruses were again much more inhibited than RES viruses. Consistent with the fusion assays, resistance to CXCL12 was more pronounced with naïve and Tcm cells than with effector cells.

This observation that RES and SENS viruses have similar cellular tropism strongly suggests that RES viruses have not emerged as a consequence of changes in the relative proportions of

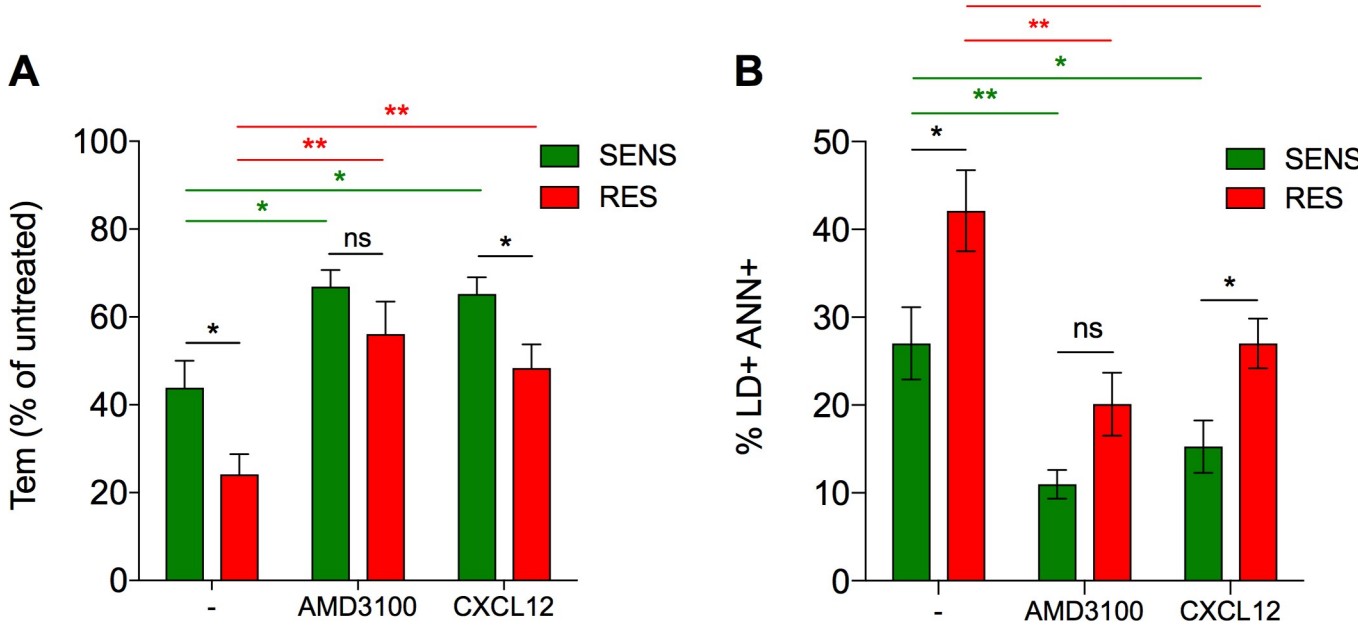

**Fig 6. Death of bystander CD4TL is increased in the presence of RES viruses. A** Percent depletion of bystander Tem cells among the fraction of SSC-high CD4TL in the presence of R5X4JT cells infected with SENS or RES viruses, compared to conditions with uninfected cells, w/ or w/o 10 μM AMD3100 or 300 nM CXCL12. **B** Percentage of Live/Dead and Annexin V positive cells within the fraction of SSC-high CD4TL after incubation with R5X4JT cells, in the presence or absence of AMD3100 or CXCL12. Results are means ± SEM of two independent experiments.

T-cell subsets during infection. We therefore investigated whether RES viruses have increased cytopathogenicity and contribute to CD4TL depletion. In preliminary experiments on activated CD4TL, we observed that viruses mainly kill uninfected "bystander" cells, but much less productively infected cells (**S7 Fig**). This is in line with literature on CD4TL depletion in HIV infection, which largely depends upon the loss of bystander cells [52,53]. We thus set up a protocol to directly compare SENS and RES viruses in their capacity to induce the death of bystander CD4TL. We infected R5X4JT cells with comparable amounts of either virus (**S8A Fig**), and then used them as effector cells in co-culture with CellTracker-labeled CD4TL (**S8B Fig**). CD4TL were then analyzed by flow cytometry for their composition in naïve and memory cells and expression of cell death markers (Live/Dead and Annexin V) (**S8C Fig**). This experiment allowed identifying a fraction of CD4TL with an effector memory phenotype that was selectively depleted in the presence of infected-R5X4JT cells. These cells were two-fold more depleted in the presence of R5X4JT cells infected with RES viruses than with SENS viruses (5-fold *vs* 2.5-fold, compared to control conditions with uninfected R5X4JT cells) (**Figs 6A and S8C**). The proportion of the remaining cells that are positive for cell death markers was also greater with RES viruses (**Figs 6B and S8C**). These results thus indicate that RES viruses more effectively kill bystander CD4TL than SENS viruses.

The CD4TL that were sensitive to the co-culture are CD4TL with high granularity (*i.e.* SSC-high cells), which most likely represent cells that are more activated than SSC-low CD4TL (**S8C Fig**). Indeed, SSC increases with expression level of the activation marker HLA-DR (**S8D Fig**) and the proportions of Tem and Temra cells increase at the expense of Tn cells in SSC-high cells, compared to SSC-low cells (**S8E Fig**). Then, we analyzed whether the death of bystander SSC-high CD4TL is dependent on Env-CXCR4 interactions. ADM3100 reversed the effects of viruses, but only partly (**Fig 6A and 6B**), suggesting that killing of bystander CD4TL occurs through mechanisms that are both dependent and independent of Env/

coreceptor interactions, as previously shown [52,53]. Of note, the differences between SENS and RES viruses were statistically abolished in the presence of AMD3100, suggesting that Env/ CXCR4 interactions are responsible for the greater ability of RES viruses to induce the death of CD4TL. Finally, CXCL12 also diminished depletion of Tem cells and expression of cell death markers, but less efficiently in the case of RES viruses. Therefore, bystander CD4TL die more in the presence of RES viruses, and they are also less protected from these effects by CXCL12.

## Discussion

CXCR4-using viruses have long been considered a poor prognosis in HIV+ patients [7,8]. However, we document here that heterogeneity exists in the pathogenicity of these viruses, which may explain why in some patients their presence does not translate into accelerated progression to AIDS [20,21]. Our data actually show that CXCR4-using viruses can be distinguished on the basis of their resistance to CXCL12, and that RES viruses are frequent in patients who experience severe depletion in peripheral blood CD4TL. CXCL12 insensitive variants had already been reported in HIV+ children, especially in children with rapidly progressing disease [54], however the mechanisms of resistance and its influence on disease progression were not studied. Our results here raise the question as to whether the presence of RES viruses is a consequence or a cause of CD4TL depletion. Our data actually suggest that RES viruses have not emerged as a result of the changes in the nature and relative proportions of T-cell subsets that come alongside with the overall decline in CD4TL levels in HIV-1 infection [55]. This is indeed unlikely because RES viruses and SENS viruses have the same tropism for T-cell subsets (**Fig 5A and 5C**). We actually propose that RES viruses could take part in deregulation of CD4TL homeostasis. CD4TL depletion is largely due to death of bystander cells, in particular through contacts with infected cells involving Env/receptor interactions [53]. As compared to SENS viruses, RES viruses more effectively deplete bystander Tem cells and increase the proportion of cells that exhibit markers for cell death (**Fig 6A and 6B**). These effects are mostly observed in a fraction of CD4TL with high granularity, which is characteristic of CD4TL that express the activation marker HLA-DR (**S8 Fig**). In the setting of chronic immune stimulation in HIV infection, the proportion of activated HLA-DR+ CD4TL, in particular within the memory subset, increases in patients [56]. These cells exhibit enhanced susceptibility to apoptosis [52], explaining our observations that they are more prone to HIV-induced cell death. The higher proportion of HLA-DR+ CD4TL in patients could also magnify the effects of RES viruses on overall Tem cell depletion. Exaggerated depletion of Tem cells could ultimately deplete the pools of Tcm and Tn cells and undermine the entire homeostasis of CD4TL [14,57], explaining the striking link between the presence of RES viruses and low CD4TL levels in patients.

CXCL12 is present in the BM [58], the thymus [59], and SLOs (lymph nodes, tonsils) [29,60]. Infection of naïve cells and of their cell precursors (CD4+ thymocytes, CD34+ multipotent hematopoietic progenitor cells) within these organs has been put forward to explain why X4 viruses deplete CD4TL more than R5 viruses [17–19,61]. In the thymus, CXCL12 is expressed by medullar epithelial cells but is also present in the cortex [59] where forming thymocytes localize and are targeted by X4 HIV-1 [62]. In SLOs, CXCL12 is abundant in the paracortical zone [29], which coincides with the region where HIV promotes depletion of bystander quiescent CD4TL by pyroptosis [61]. These quiescent cells can represent over 95% of the CD4TL in SLOs, and their death by pyroptosis is a major driver of immunodeficiency in HIV infection. Importantly, blockade of Env-CXCR4 interactions with AMD3100 prevents bystander CD4TL death [15]. CXCL12 in the immediate environment of bystander CD4TL could theoretically do the same, however, we show here that this will depend on sensitivity of

viruses to the chemokine. CXCL12 is actually strongly limited in its capacity to block entry of RES viruses into CD4TL, in particular into naïve cells (**Fig 5**), which represent a great part of the quiescent cells in SLOs. This suggests that RES viruses will induce more effectively pyroptosis of quiescent CD4TL than SENS viruses. For the same reasons, resistance to CXCL12 could also lead RES viruses to be more effective than SENS viruses in targeting CD4TL precursors in the thymus and the BM. Overall, RES viruses are likely to cause a more massive depletion of naïve CD4TL than SENS viruses, thereby contributing more to the overall reduction of circulating CD4TL numbers in patients.

Our results provide clues to diagnose highly pathogenic X4 viruses in patients. Resistance to CXCL12 is associated with changes in the epitope of V2-apex bNAbs, and in the case of F8, at the gp120/gp41 interface (**Fig 3**). Whether these alterations directly contribute to resistance to CXCL12, or are the consequences of resistance mutations in other Env regions, was not assessed here. Of interest however, both regions were shown to regulate opening of the Env trimer [41,42]. We therefore propose that the amino acid changes in these regions of RES Envs cause them to adopt a more closed conformation, compared to SENS Envs, explaining why they are more resistant to sCD4 (**Fig 2C and 2E**). It should be noted that RES Envs from the ACS and F8 Env however are differentially sensitive to V2-apex bNAbs (e.g. PGT145) and PGT151 (targeting the gp120/gp41 interface) (**Fig 3A and 3B**), and recent single-molecule FRET experiments have shown that V2-apex bNAbs and PGT151 do not recognize the same conformations of the Env trimer [63]. Altogether, these data suggest that F8 Env does not adopt the same conformation as RES Envs from the ACS, and that distinct conformational states of the Env trimer may be resistant to CXCL12.

Another peculiarity of RES Envs is that they do not interact with the same CXCR4 molecules as SENS Envs. In this regard, our results add to an ever-growing literature on heterogeneity of G protein coupled receptors [64]. They extend previous observations showing that CXCR4 adopts distinct antigenic conformations that can be differentially used by distinct viruses [48,49]. The fact that RES and SENS Envs do not bind the same CXCR4 molecules also suggests that they do not activate the same intracellular signaling pathways. Activation of the pathways involved in programmed cell death could explain why RES Envs more effectively kill bystander CD4TL (**Fig 6**).

Here, we further demonstrate that CXCR4 heterogeneity is the basis of resistance to CXCL12. Contrary to what we hypothesized first, resistance is not due to increased affinity of X4 gp120s for CXCR4, which remains to a large extent lower than that of R5 gp120/CCR5 interactions (Kd > 100 *vs* 10 nM) [32], but is related to usage of CXCR4 receptors that are weakly or not accessible to CXCL12. These results are reminiscent of the mechanisms by which R5 HIV-1 strains escape inhibition by CCR5 chemokines [26]. In contrast to CXCR4-using viruses, however, R5 viruses escape chemokines regardless of the stage of infection and the immunological status of patients (**Figs 1B and** S1C and S1D). Previous work reported increased resistance to CXCR4 antagonists (e.g. AMD3100) for late X4 viruses (including some isolated from P#39 and P#208 of the ACS) [65]. However, we did not necessarily reproduce this observation, suggesting that the CXCR4 molecules that do not bind CXCL12 may preserve interaction with other ligands. In line with this, our results in **Fig 4** strongly suggest that binding to CXCR4 molecules not accessible to CXCL12 is the reason why mAb 4G10 efficiently inhibits RES viruses.

Our observation that RES viruses bind to receptors that differ from those to which CXCL12 binds raises new perspectives for the treatment of HIV+ patients. Historically, the development of HIV-1 entry inhibitors targeting CXCR4 has been limited due to the potential adverse effects that could result from blocking the homeostatic functions of the receptor [22]. Our results here suggest that RES viruses are more prone to deregulate CD4TL homeostasis

compared to other CXCR4-using viruses. Therefore, as part of a cART regimen in HIV + patients, we propose that pharmacological blocking of the interactions between RES viruses with their specific receptors could help prevent CD4TL depletion and progression to the disease. Because these receptors bind CXCL12 very loosely, if at all, it is unlikely that this strategy impairs the normal functions of the chemokine. An ideal drug candidate should thus bind low-CXCL12 affinity receptors, but not high-CXCL12 affinity receptors. Screening of molecule libraries against RES and SENS viruses and identifying the molecular determinants that distinguish the low- and high-CXCL12 affinity receptors, might help find out such a molecule. In this regard, the lipid environment of CXCR4 may play a role. Indeed, previous work showed that CXCR4 can exist in two distinct conformations depending on its location in cholesterol-rich and cholesterol-poor regions, one binding CXCL12 and the other not, respectively [66]. But other factors that are at work to contribute to CXCR4 heterogeneity (e.g. its association with membrane or cytosolic proteins, Tyrosine sulfation) [67–69] could also influence CXCL12 binding.

Finally, this work sheds new light into the role of CXCL12 in HIV-1 transmission. CXCL12 within mucosa and lymph nodes was proposed to counter-select transmission of CXCR4-using viruses [28,29], thereby contributing to preponderance of R5 viruses in the first stages of HIV-1 infection. We thus anticipated that resistance to CXCL12 could be a prerequisite for transmission of viruses using CXCR4. However, resistance to CXCL12 is rare among recently transmitted viruses that use CXCR4, regardless of the mode of transmission (parenterally or sexually) (**Fig 1D** and S4 Table), suggesting that factors other than CXCL12 contribute to counter-selection of CXCR4-using viruses. Studying these factors could provide novel insights into HIV-1 transmission and ways to prevent it.

## Materials and methods

### Ethics statement

The CD4TL used in this study have been isolated from blood samples purchased at Etablissement Français du Sang (EFS, the French National Blood Agency). Healthy blood donors from EFS were anonymous and provided written informed consent to EFS at the time of blood collection. The biological virus clones from PBMCs of participants of the Amsterdam Cohort Studies (ACS) on HIV-1 and AIDS have been provided to us by Pr H. Schuitemaker, University of Amsterdam, The Netherlands. The ACS (https://www.amsterdamcohortstudies.org) was created in 1984 and enrolled HIV-1 infected individuals and HIV-1-uninfected individuals at high risk for HIV infection among homosexual men and drug users. The ACS obeys the principles expressed in the Declaration of Helsinki and was approved by the Medical Ethics Committee of the Academic Medical Center in Amsterdam. Participants were anonymous volunteers who provided written informed consents for blood usage in research investigations. Eight of the nine individuals who were diagnosed at an advanced stage of infection, to whom reference is made in **Figs 1C** and S1E and S1F, are anonymous participants of the ANRS 146 OPTIMAL trial (Number of registration in http://clinicaltrials.gov: NCT01348308) (see also **S3 Table** for details). The objective of this study was to investigate the influence of maraviroc adjunction to cART in immune recovery and disease progression in treatment naïve and HIV-1-infected adults with CD4TL count below 200/μl and/or an AIDS-defining illness. The blood samples, from which the envelope glycoproteins studied here were isolated, were drawn at the time of inclusion. Patients gave written consent for conservation and use of blood to research purposes. The ninth patient diagnosed late in infection, as well as those diagnosed at the chronic stage (in whom the chronic Envs in **Fig 1C** were isolated) or at the time of PHI and harboring CXCR4-using Envs (**Fig 1D**, with the exception of Envs cl.2.2 and cl.19.1) are part

of a cohort of more than 3,000 HIV-1-infected patients who attended the Department of Infectious Diseases of Toulouse University Hospital between 1995 and 2018. All patients here provided written informed consent for conservation and handling in research activities (e.g. tropism and coreceptor usage assays) of their blood samples drawn at the time of their diagnosis. In accordance with recommendations of the National Commission for data protection and liberties (CNIL), they also consented to anonymous exploitation of their biological (e.g. CD4TL, viral load) and clinical data. The individuals diagnosed at the time of PHI infected with Envs cl.2.2, cl.19.1 (**Fig 1D**) or the R5 Envs shown in **S1C and S1D Fig** are patients who attended the Sandoval Health Center (Madrid, Spain). The project and the informed consent were approved by the Ethical Comittee of Instituto de Salud Carlos III (PI 17_2013-V5).

## Patient characteristics

The available baseline characteristics of the patients of the ACS infected with CXCR4-using or R5 viruses, of those referred to in **Fig 1C**, and of the patients diagnosed at the time of PHI (**Figs 1D and** S1C and S1D) are shown in **S1–S4 Tables**, respectively. The Patients of the ACS from whom R5 Envs were isolated had previously been identified as Patients H1 (P#1), H2 (P#458), H3 (P#1031), H4 (P#232) and H5 (P#341) [70,71]. Patients #39, #208 and #1120 have already been described in the following references [65,72]. Viral tropism was determined phenotypically using both U87-CD4 cells expressing CXCR4 or CCR5 and R5X4JT cells in the presence of MVC and/or AMD3100, with similar results. Recent HIV-1 seroconversion of the patients depicted in **S4 Table** was defined according to Fiebig et al. [73]. In contrast to the other recently infected patients, Patient F8 exhibited a severe depletion in blood CD4TL at diagnosis (CD4TL count = 45/mm$^3$ and CD4/CD8 T cell ratio = 0.08). The patient received combined antiretroviral therapy with 2 nucleoside reverse transcriptase inhibitors and boosted protease inhibitor. After 12 months of effective cART and suppression of plasma viremia (1.3 log$_{10}$ copies/ml), immune reconstitution of patient F8 remained weak (CD4+ T-cell count = 147/mm$^3$ and CD4/CD8 ratio = 0.25).

## Cells

U87-CD4, U87-CD4-CCR5 and U87-CD4-CXCR4 cells were obtained from the NIH AIDS Research and Reference Reagent Program (NARRRP). U87-CD4-CXCR4 cells were subcloned to obtain a new cell line with improved detection sensitivity to infection, which was used here and described previously [74]. These cells were cultured in Dulbecco's Modified Eagle Medium (DMEM) supplemented with 10% (v/v) foetal bovine serum (FBS), 300 μg/ml geneticin, 10 mM HEPES, 1 μg/ml puromycin (U87-CD4-CCR5/CXCR4 only) and 2 μg/ml blasticidin (U87-CD4-CXCR4 only). HEK 293T cells were obtained from ATCC. These cells served to produce the CCR5- (HEK-R5 cells) and CD4- (HEK-CD4) expressing HEK 293T cells, which were described previously [32]. TZM-bl cells were obtained from the NARRRP and were maintained in Dulbecco's Modified Eagle Medium (DMEM) supplemented with 10% FBS, 100 μg/ml streptomycin and 100 units/ml penicillin. BHK-21 cells used for production of gp120s were obtained from ATCC and were cultured in Glasgow's Minimum Essential Medium (GMEM) supplemented with 5% FBS, 10% Tryptose Phosphate Broth, 2% HEPES and streptomycin and penicillin. The Jurkat T cell line expressing CCR5 (R5X4JT cells, obtained from Dr F. Bachelerie, Institut Pasteur, Paris) was previously described [75] and cultured in RMPI 1640 supplemented with 10% FBS, streptomycin and penicillin. Human CD4TL were purified from PBMCs of healthy blood donors from EFS as described previously [32]. They were maintained for 2 days in complete RPMI 1640 medium containing

recombinant interleukin-2 (IL-2) (300 IU/ml) and phytohemagglutinin (PHA) (1 μg/ml, Thermo Fisher Scientific), and then for additional 3 days in the presence of IL-2 alone.

## Chemicals and antibodies

Recombinant sCD4 was produced and purified as previously described [76]. MVC and AMD3100 were obtained from Sigma-Aldrich (cat# PZ0002 and A5602, respectively) and CXCL12 from Miltenyi Biotec (cat# 130-093-998). P2G was provided by Dr F. Baleux (Institut Pasteur, Paris). The following antibodies were used for membrane or intracellular staining: CD45RA-APC (clone T6D11), CCR7-PercP-Vio700 (clone REA546), HLA-DR-APC-Vio770 (clone AC122), CD3-PE-Vio615 (clone REA613), CCR5-PE-Vio770 (clone REA245), CXCR4-PE (clone REA649) (all from Miltenyi Biotec); CD4-BV786 (clone SK3) (from BD Biosciences); CXCR4 (clone 4G10) (Santa Cruz Biotechnology); Unlabeled CXCR4 mAb clones (12G5, 44716, 44717, 44708) (all from R&D Systems); goat anti-mouse secondary antibody-AF647 (from Invitrogen); HIV-1 core antigen-FITC (clone KC57) (from Beckman Coulter). The anti-CD4 neutralizing mAb Q4120 was provided by Dr. Q. Sattentau and the NIBSC Centralised Facility for AIDS Reagents. The broadly neutralizing HIV-1 antibodies used here (PGT145, PG16, PGDM1400, PGT151, PGT121, VRC01, 10E8) were produced as recombinant human IgG1 monoclonal antibodies by co-transfection of Freestyle 293-F suspension cells (Thermo Fisher Scientific) as previously described [77]. Antibodies were purified by batch/gravity-flow affinity chromatography using protein G-coupled beads according to the manufacturer's instructions (Protein G Sepharose 4 Fast Flow, GE Healthcare). Purified antibodies were dialyzed against PBS, and IgG concentration was measured using the NanoDrop One instrument (Thermo Fisher Scientific).

## Description and cloning of env sequences and production of viruses

The nucleotide sequences of the CXCR4-using gp120s or gp160s derived from the individuals of the ACS are available in GenBank (see Data Availability Statement for details). The R5 Envs of the ACS have previously been described by us [32] and others [70,71]. The identification numbers of the virus biological clones and the patients from whom the glycoproteins were isolated are shown in **S1 and S2 Tables**. The sequences of gp120s and gp160s were cloned in the pSFV2 expression vector and the pNL-SacII-lacZ/env-Ren proviral vector, respectively, as described [30,32]. For preparation of recombinant virus populations derived from plasma of HIV+ patients, the region of the *env* sequence that encompasses gp120 and the ectodomain of gp41 (*i.e.* gp140) was amplified by RT-PCR from the viral RNA and then submitted to a nested PCR, as detailed previously [74]. The PCR product was then cotransfected with NheI-linearized pNL43-Δenv-Luc2 vector DNA in HEK 293T cells using the calcium-phosphate DNA coprecipitation method [74]. The virus clones pseudotyped with the Envs of the ACS were also produced in HEK 293T cells but using polyethylenimine as transfection agent [78]. Forty-eight hours post-transfection, cell culture supernatants were harvested and quantified for their Gag p24 content by ELISA (Innotest HIV antigen mAb; Innogenetics, Gent, Belgium). Viruses were then stored at– 80˚C until they were used.

## Cell sorting

Freshly purified CD4TL were FACS-sorted after their treatment with PHA/IL-2 for 5 days, as described above. Then, cells (10 x $10^6$/ml) in staining buffer (PBS, 1% human serum) were incubated with PerCP-Vio 700-conjugated anti-human CCR7 mAb (1:20 dilution, Miltenyi Biotec) and APC-conjugated anti-human CD45RA mAb (1:100, Miltenyi Biotec) for 20 min at 4˚C and filtered through 70 μm cell strainers. Sorting of Tn (CCR7+, CD45RA+), Tcm

(CD45RA-, CCR7+), Tem (CCR7-, CD45RA-) and Temra (CCR7-, CD45RA+) cells was then performed using the FACSARIA-SORP cell sorter (BD Biosciences). The CD4TL subsets were then immediately infected after sorting.

## Infection assays

Infectivity of Env-pseudotyped viruses was determined in single cycle infection experiments carried out as follows. U87 cells in flat bottom 96-well plates ($5 \times 10^3$ cells/well) or CD4TL treated with PHA/IL-2, FACS-sorted CD4TL subsets, or R5X4JT cells in round bottom 96-well plates ($2 \times 10^5$ cells/well) were incubated for 48 h at 37°C in complete DMEM (U87) or RPMI (lymphocytes) medium (supplemented with 20 ng/ml IL-2 in the case of primary CD4TL) with 20 ng Gag p24 of viruses. Cells were then lysed and infectivity was determined by measuring Renilla or Firefly luciferase activity, as appropriate (Renilla Luciferase Assay System or Luciferase assay System, Promega, Madison, WI, USA). For infection inhibition with compounds binding CD4 (Q4120) or coreceptors (chemokines, MVC, AMD3100 and anti-CXCR4 mAbs), target cells were incubated for 20 min with the inhibitors alone prior addition of viruses. In the case of dose-response inhibitions by chemokines or Q4120, experiments were typically performed using three-fold serial dilutions of inhibitors (between $10^{-6}$–$10^{-10}$ M for chemokines, and $3 \times 10^{-8}$–$10^{-12}$ M for Q4120). For inhibition assays with sCD4, viruses were incubated with eight different inhibitor concentrations ($10^{-6}$–$10^{-10}$ M) for 90 min at 37°C. Then, the virus-sCD4 mixtures were added to CD4TL for additional 48 h. Neutralization sensitivity of Nef-positive and Luc-negative Env-pseudotyped viruses to bNAbs was determined using TZM-bl cells. Equal amounts of viruses (100,000 RLU equivalents) were incubated in 96-well plates with four-fold serial dilutions of concentrations of bNAbs (7 points, 25–$1.5 \times 10^{-3}$ μg/ml) for 1 h at 37°C. Then, TZM-bl cells ($1 \times 10^4$) were added to the antibody-virus mixtures and incubation was continued for 48 h. Infectivity was then determined by measuring the level of Tat-induced luciferase expression in the lysates of TZM-bl cells. Uninfected cells were used as control of background luciferase signal. The percent inhibition of infectivity was calculated relative to control infection measured in the absence of antibody. In all the neutralization assays described, results were fitted to a sigmoidal dose-response model with a variable slope. $IC_{50}$ values were determined from inhibition curves with GraphPad prism 6.

## Virus-cell fusion assays

Viruses containing the BLaM-vpr fusion protein have been prepared in HEK 293T cells as previously described [32], except that PEI was used as transfection reagent in place of calcium phosphate. One hundred ng p24 of viruses (unless otherwise specified) were then exposed to $2 \times 10^5$ CD4TL in the presence or absence of inhibitors (CXCL12, AMD3100, MVC) by spinoculation at 4°C (1 h, 2,000 $g$). The virus-cell mixtures were then warmed to 37°C for 2 h. Cells were then washed twice at RT and further incubated with the CCF2/AM dye for 2 h at 37°C according to the manufacturer's instructions (Invitrogen). In some experiments, CCF2-loaded cells were further stained for 30 min at RT with the 605 nm emitting LIVE/DEAD viability dye (Invitrogen) and then at 4°C for 30 min with PerCP-Vio 700-conjugated anti-human CCR7 mAb, APC-conjugated anti-human CD45RA mAb (Miltenyi Biotec) and BV786 anti-human CD4, clone SK3 (BD Biosciences). Cells were then fixed with 2% PFA in PBS for 20 min at 4°C. The percentage of cells with CCF2 cleaved by BLaM was quantified by flow cytometry using the BD LSR-Fortessa (BD Biosciences).

## Radioligand and antibody binding experiments

The crude membrane preparations used in the gp120 binding experiments were obtained from newly purified PBMCs from healthy donors or CCR5-expressing HEK 293T cells as described previously [79]. The protocol for production of $^{35}$S-labeled gp120s has also been detailed elsewhere [30]. The binding experiments to CCR5 and CD4 were carried out similarly as in our recent works ([32] and [30], respectively). Equilibrium saturation binding of X4 $^{35}$S-gp120 was performed in Eppendorf tubes as follows. Total binding (TB) of increasing concentrations of gp120s (10–140 nM) to 40 μg of membrane proteins was determined in 0.1 ml of assay buffer (50 mM HEPES, pH 7.4, 1 mM CaCl$_2$, 5 mM MgCl$_2$, 5% BSA) containing 400 nM sCD4. Incubations were carried out for 90 min at RT. Non-specific binding (NSB) of gp120s was measured in the presence of 10 μM AMD3100. Under these conditions, NSB did not represent more than 30% of TB and did not vary with quantity of PBMC membrane, as expected for NSB, ruling out that a fraction of $^{35}$S-gp120 has bound to membrane CD4. Unbound gp120s were then removed by two brief high-speed centrifugations (16,000 $g$ in an Eppendorf 5415 R centrifuge), separated by a washing step in ice-cold washing buffer (50 mM HEPES, pH 7.4, 1 mM CaCl$_2$, 5 mM MgCl$_2$, 400 mM NaCl). Membrane pellets were then resuspended in scintillation liquid (PerkinElmer Life Sciences) and radioactivity was counted in a Wallac 1450 MicroBeta TriLux (PerkinElmer Life Sciences). Specific binding (SB) of $^{35}$S-gp120/sCD4 complexes to CXCR4 was then obtained by subtracting from TB the NSB. Fitting of data points to a one-site binding model and calculation of equilibrium dissociation constants (Kd) and maximum levels of binding (Bmax) were performed using GraphPad Prism 6. The displacement experiments of $^{35}$S-gp120 binding by CXCL12 were carried out using the same protocol as just described above. Results were normalized for NSB (0%) and SB in the absence of CXCL12 (100%) and were fitted to a sigmoidal dose-response model with a variable slope. The binding experiments of $^{125}$I-CXCL12 (PerkinElmer) to PBMC membranes were performed in Minisorp tubes (Nunc, Rochester, NY) as follows. In 0.1 ml final volume of assay buffer (50 mM HEPES, pH 7.4, 1 mM CaCl$_2$, 5 mM MgCl$_2$, 0.5% BSA), 0.5 nM of the chemokine was incubated for 90 min at RT with 15 μg of membrane proteins in the presence or absence of 10 μM AMD3100 (NSB) and/or 100 μM Gpp(NH)p. Membranes were then filtered through 1% PEI-pretreated GF/B filters and washed three times with washing buffer (the same as above). The radioactivity remaining bound to filters was then counted in a gamma counter (multi-crystal LB 2111 gamma counter, Berthold Technologies).

For the saturation binding experiments of anti-CXCR4 mAbs to CD4TL, cells (1 x 10$^5$) in conical 96-well plates were incubated for 2 h at RT with increasing concentrations of mAbs (0.025–10 μg/ml) in 0.15 ml PBS supplemented with 1% human serum. They were then washed once with ice-cold PBS and incubated for additional 30 min at 4˚C with AlexaFluor 647-conjugated goat anti-mouse (GAM) IgG (Invitrogen) used at a 1:500 dilution. After a last washing step at 4˚C, geometric mean fluorescence intensities of cells as indicators of mAb binding were then analyzed by flow cytometry using the BD LSR-Fortessa (BD Biosciences). Results were subtracted from background signal of GAM IgG alone and fitted to a one-site binding model using GraphPad Prism 6. The competition assays of anti-CXCR4 mAb binding by CXCL12 were performed using the following protocol. Cells (1 x 10$^5$ CD4TL, or 5 x 10$^4$ U87 cells in suspension after being detached in PBS containing 2 mM EDTA) were incubated for 2 h at RT in assay buffer (PBS; 0.2% BSA; 0.1% NaN$_3$) containing a constant concentration of anti-CXCR4 mAb (equal to 3-fold its Kd) and increasing concentrations of unlabeled CXCL12. As above, mAb binding was then revealed by AF 647-conjugated GAM IgG staining and flow cytometry analysis. Results were fitted to one-site competitive binding model using GraphPad Prism 6. From the competition curves, the affinity constants Ki of CXCL12 were

then calculated according to the Cheng and Prusoff equation [80]: $Ki = [IC_{50}/(1+[mAb]/Kd)]$, where Kd and [mAb] represent the affinity (from **S5 Fig**) and the concentration of the tested mAb, respectively.

## HIV-1 Env-induced cell death assays

The Env-expressing R5X4JT cells that were used in the cell death assays of CD4TL were prepared as follows. VSVG-pseudotyped viruses were first produced in HEK 293T cells by cotransfection of SENS or RES Env-containing pNL-SacII-lacZ/env-Ren proviral vector and pVSVG using the PEI-based transfection method. Viruses were then harvested 48 h later and used to infect R5X4JT cells for 2 days. Virus inocula were adjusted so that the percentage of infected R5X4JT cells was approximately 20%, as controlled by quantification of p24-positive cells by flow cytometry. Infected R5X4JT cells were then incubated with PHA/IL-2-activated CD4TL pre-loaded with CellTracker (7-amino-4-chloromethylcoumarin [CMAC]) dye (Life Technologies) (25 µM CellTracker were mixed with 10 x $10^6$ cells/ml of RPMI 1640 medium for 20 min at 37˚C). The co-cultures were run for 2 days with 5 x $10^5$ of each cell type in 500 µl of assay medium (complete RPMI 1640 medium supplemented with 300 IU/ml IL-2) / well of 24-well plates. Cells were washed with PBS, stained for 30 min at RT with the Invitrogen LIVE/DEAD fixable dead cell staining kit and then at 4˚C for 30 min with PerCP-Vio 700-conjugated anti-human CCR7 mAb and APC-conjugated anti-human CD45RA mAb (Miltenyi Biotec). Cells were then washed with Annexin V binding buffer and stained with Annexin V conjugated to PE (Miltenyi Biotec). Finally, expression levels of Gag p24 in the cells were assessed by intracellular staining with KC57 FITC-conjugated antibody (Bechman Coulter) (45 min at RT) after treatment with the BD Cytofix/Cytoperm kit. Staining of cells was then analyzed by flow cytometry using the BD LSR-Fortessa (BD Biosciences).

## Statistical analysis

Curve fitting and determination of $IC_{50}$ values were carried out using GraphPad Prism 6. Most data, unless otherwise stated, are representative of at least three independent experiments carried out in duplicate or triplicate. The independent infection experiments on primary CD4TL or PBMCs were performed using samples from distinct healthy blood donors. If not mentioned otherwise, statistics were run using the non-parametric Mann-Whitney U-test. *P* values are presented directly in the figures as follows: ns, $P > .05$ (not significant); *, $P < .05$; **, $P < .01$; ***, $P < .001$; ****, $P < .0001$.

## Supporting information

**S1 Fig. CXCR4-using viruses, but not R5 viruses, exhibit differences in their sensitivity to chemokines in the course of HIV-1 infection (Related to Fig 1). A** and **B** Inhibition by CXCL12 of early (viruses # 1 and 16) and late (viruses # 6 and 28) Envs from PBMCs of two patients of the ACS (P#39 (**A**) and P#208 (**B**)) (Related to **Fig 1A**). Data points (means ± SEM of triplicate determinations) are expressed as percent infection of CD4TL relative to control infection measured in the absence of CXCL12 (100%) and were fitted to a sigmoidal dose-response model with a variable slope. $IC_{50}$s of CXCL12 were derived from inhibition curves using GraphPad Prism 6. Representative experiments out of at least three independent experiments carried out on CD4TL from distinct healthy donors are shown. **C** and **D** Percent infection of PHA/IL-2-activated PBMCs from healthy donors with 20 ng p24 of viruses pseudotyped with Envs isolated at the stage of PHI or the early or late R5 Envs shown in **Fig 1B**, in the presence of 20 nM CCL3 (**C**) or 10 nM CCL5 (**D**). Each data point represents the mean infection measured for a given virus (n = 3 determinations), expressed relative to

infection in the absence of chemokine (100%). One (**C**) and two (**D**) out of three independent experiments are shown. Error bars represent the SD to the means. ns, not significant; \*\*P < .01 in the Mann-Whitney test. **E** Coreceptor usage of recombinant virus populations pseudo-typed with the Envs depicted in **Fig 1C**, isolated at the time of diagnosis, at the chronic or late stage of infection. Results (means ± SEM of triplicate determinations) represent infectivity of viruses measured 48 h post-inoculation of R5X4JT cells (with 20 ng of p24), in the presence or absence (black bars, 100%) of 10 μM maraviroc (MVC, light green bars), 10 μM AMD3100 (AMD, orange bars) or a mixture of both antagonists at 10 μM each (pink bars). A representative experiment out of three is shown. Group-1 and group-2 refer to late Envs that are not, or are, significantly more resistant to inhibition by CXCL12, as compared to chronic Envs and NL4-3. **F** Correlation analysis (Spearman two-tailed test) between the $IC_{50}$s of CXCL12 for inhibiting the plasma-derived Envs shown in **Fig 1C** and the CD4TL count in the blood of patients. Error bars are means ± SEM of 2 to 3 independent determinations. ns, not significant. **G** Coreceptor usage of recombinant viruses pseudotyped with the Envs depicted in **Fig 1D**, isolated at the stage of PHI. Results (means ± SEM of triplicate determinations) represent infectivity of viruses measured 48 h post-inoculation of activated CD4TL (with 20 ng of p24), in the presence or absence (black bars, 100%) of 10 μM maraviroc (MVC, light green bars), 10 μM AMD3100 (AMD, orange bars) or a mixture of both antagonists at 10 μM each (pink bars). Luciferase activity in the lysates of infected cells, expressed as relative light units (RLU) was used to quantify virus infectivity. One representative experiment out of three is shown. **H** Inhibition by CXCL12 of recombinant viruses pseudotyped with the indicated Envs isolated at the time of PHI. Data points (means ± SEM of triplicate determinations) are expressed as percent infection of activated CD4TL relative to control infection measured in the absence of CXCL12 (100%) and were fitted to a sigmoidal dose-response model with a variable slope. $IC_{50}$s were calculated using Prism 6. A representative experiment out of three independent experiments carried out on CD4TL from distinct healthy donors is shown. **I** and **J** Inhibition by CXCL12 of infection of CD4TL with viruses pseudotyped with chimeric Envs containing gp120 from the early (#1 and #16) or late (#6 and #28) viruses of P#39 (**A**) and P#208 (**B**), combined with NL4-3 gp41. Equal amounts of viruses were used (50 ng of Gag p24). Inhibition curves were fitted according to a sigmoidal dose-response model with a variable slope. Data points (means ± SEM of triplicate determinations) are expressed as percent infection of CD4TL relative to control infection measured in the absence of CXCL12 (100%). Representative experiments out of three independent experiments carried out on CD4TL from distinct donors are shown. **K** Fusion of chimeric viruses containing late gp120s (red bars) with CD4TL is more resistant to CXCL12 inhibition, compared with viruses with early gp120s (green bars). βLactamase (BLaM)-Vpr-containing viruses (500 ng of Gag p24) were incubated with CD4TL loaded with the fluorescent BLaM substrate CCF2. Fusion was quantified by flow cytometry by counting the number of cells with cleaved CCF2. Results are expressed as percent fusion in the presence of 30 or 300 nM CXCL12 relative to control fusion measured in the absence of the chemokine (100%). Shown are the means ± SEM of two independent experiments carried out with the viruses containing gp120s #1 (early), #6 (late), #16 (early) and #28 (late). \*\*$P$ < .01, Mann-Whitney test.
(PPTX)

**S2 Fig. Analysis of Env incorporation into viruses pseudotyped with CXCL12-sensitive and CXCL12-resistant Envs. A** Western blot analysis of gp120 and p24 expression into viruses pseudotyped with early (SENS) Envs (#1, 16, 29, 36 and 44), late (RES) Envs (#6, 28, 32, 41 and 47) or NL4-3 Env. Three hundred ng of Gag p24 of the different viruses were solubi-lized in lysis buffer containing XT sample buffer (Biorad), Invitrogen NuPAGE sample

reducing agent and 1% Triton X-100, incubated for 5 min at 70˚C and then loaded onto Biorad Criterion XT 4–12% Bis-Tris gels under reducing conditions and then transferred to nitrocellulose membrane. Membranes were blocked with Odyssey blocking buffer (Li-COR) (for p24 detection) or TBS containing 5% BSA and 0.05% NaN3 (for gp120 detection) and then incubated overnight at 4˚C with a sheep anti-HIV-1 gp120 polyclonal antibody (clone D7324, Aalto Bio Reagents) or for 1 h at RT with a mouse anti-HIV-1 p24 mAb (clone 749140, R&D Systems). Membranes were then incubated with the following species-specific secondary antibodies: DyLight 800-conjugated donkey Anti-Sheep IgG (Novusbio) and IRDye 800CW-conjugated goat Anti-Mouse (Li-COR) (dilution: 1/10,000). Signals were detected with a Li-COR Odyssey scanner and quantified using ImageStudioLite software. Arrow indicates gp120 bands. As control, thirty ng of purified gp120 #28 were also immunodetected. A representative experiment out of two independent experiments with distinct virus preparations is shown. **B** Band intensity ratios of gp120 to p24, normalized to NL4-3. Means ± SEM of two independent experiments are shown. The color code is the same as in **Fig 1A**. Statistics: Mann-Whitney test.
(PPTX)

**S3 Fig. CXCL12-resistant Envs display amino acid changes in the epitopes for V2-apex and gp120/gp41 interface bNAbs (Related to Fig 3). A** and **B** Sensitivity of SENS, RES, F8 and F9 Envs to neutralization by PG16 (**A**) and PGDM1400 (**B**). Experiments were carried out and analyzed as described in **Fig 3**. **C** RES Envs, but not SENS Envs, show amino acid substitutions in the epitope for V2-apex bNAbs. The corresponding sequence in HxB2 Env is shown as reference. The color code for the patients is the same as in **Fig 3**. **D** Different residues that comprise the epitope for PGT151 at the gp120/gp41 interface are substituted in F8 Env.
(PPTX)

**S4 Fig. Binding of RES viruses to CXCR4 is less effectively inhibited by CXCL12, compared to SENS viruses.** The panel shows inhibition by CXCL12 or its antagonist variant P2G of infection of primary CD4TL with viruses pseudotyped with NL4-3 Env, SENS Envs (Envs #1 and #16) or RES Envs (Envs #6 and #28). Shown are the means ± SEM of at least three independent experiments carried out with CD4TL from distinct healthy donors. *, $P < .05$; **, $P < .01$; ***, $P < .001$, Mann-Whitney test.
(PPTX)

**S5 Fig. Role of G proteins in the binding of gp120s and chemokines to CCR5 and CXCR4 and identification of antigenically distinct conformations of CXCR4 on CD4TL (Related to Fig 4). A** Coupling of CCR5 to G-proteins is required for the binding of CCL3, but not for the binding of R5 gp120s. The panel represents the specific binding of $^{125}$I-CCL3 (0.1 nM) or of the R5 $^{35}$S-gp120s (10 nM in complex with 30 nM sCD4) from patients P#1, #341, #458 and #1031 to membrane preparations from CCR5-expressing HEK 293T cells, in the presence or absence of 100 μM of 5'-guanylylimidodiphosphate (Gpp(NH)p), a nonhydrolysable GTP analog that uncouples permanently G proteins from receptors. Specific binding was deduced by subtracting from total binding the non-specific binding measured in the presence of 10 μM maraviroc. A representative experiment out two is shown. Results (expressed as % binding relative to binding in the absence of Gpp(NH)p) are means ± SEM of technical triplicates. **B** In contrast to CCL3, the binding of CXCL12 (0.5 nM) or of the X4 $^{35}$S-gp120s (80 nM in complex with 300 nM sCD4) from patients P #39 and #208 to membrane preparations from PBMCs is poorly sensitive to treatment with 100 μM Gpp(NH)p. Specific binding was calculated as in panel **A**. Non specific binding of CXCR4-using ligands was determined with 10 μM AMD3100. Means ± SEM of two independent experiments are shown. **C** Saturation binding of

anti-CXCR4 mAbs to CD4TL. PHA/IL-2-treated CD4TL (1 x $10^5$ cells) in PBS supplemented with 1% human serum were incubated for 2 h at room temperature with increasing concentrations of mAbs (in µg/ml and in nM) targeting the second extracellular loop (ECL2) of CXCR4 (mAbs 44 708, 44716 and 44717), its N-terminus (Nt) (4G10) or a conformational epitope encompassing ECL2, ECL3 and the Nt (12G5). Cells were then stained with an AlexaFluor 647-conjugated goat anti-mouse IgG secondary antibody for 30 min at 4˚C. GMFI values for each mAb were determined by flow cytometry analysis, and subtracted of the GMFI obtained for the secondary antibody alone. Binding levels of mAbs were inferred from GMFI values and are expressed as percent of maximal binding of mAb 44717. Means ± SEM of two independent experiments using cells from distinct donors are shown. Fitting of results to a one-site binding model and determination of equilibrium dissociation constants Kd were done using GraphPad Prism.
(PPTX)

**S6 Fig. Characteristics of CD4TL subsets with respect to permissiveness to SENS and RES viruses and receptor expression levels (Related to Fig 5). A** Percent fusion of pseudotyped viruses with CD4TL. Cells (2 x $10^5$) were incubated with 100 ng Gag p24 of BLaM-vpr-containing viruses pseudotyped with either of the indicated Envs and then loaded with the BLaM substrate CCF2-AM. Percent fusion was determined by flow cytometry and represents the proportion of cells where CCF2 is cleaved by BLaM. For each virus, the mean percentages of fusion obtained from three independent experiments are plotted. **B** Strategy for quantifying the propensity of each CD4TL subsets to fuse with SENS and RES viruses. The different CD4TL subsets within all cells (gated in red) and within infected cells (i.e. cells positive for cleaved CCF2, gated in blue) were distinguished based on their capacity to bind anti-CD45RA and anti-CCR7 mAbs. This allowed calculating the PR value for each cell subset, which represents the ratio of its proportions in infected *vs* all cells. In this example, PR = 1.2 for Tcm cells, indicating that they are more abundant in infected than in uninfected cells. **C** Geometric Mean Fluorescence Intensity (GMFI) of CXCR4, CCR5 and CD4 mAbs on the different CD4TL subsets. CD4TL were activated for 5 days with PHA/IL-2 and then FACS-sorted prior to analysis of receptor expression levels. **D** Percent cells that express CXCR4, CCR5 or CD4 on each CD4TL subset.
(PPTX)

**S7 Fig. HIV-1 infection of CD4TL induces the death of bystander cells but only moderately the death of productively infected cells (Related to Fig 6).** PHA/IL-2-treated CD4TL (2 x $10^5$ cells) were incubated with 50 ng p24 of SENS or RES viruses for 7 days. Productively infected (p24 positive) and bystander (p24 negative) CD4TL were then stained with the Live/Dead viability dye (L/D) and analyzed by flow cytometry. A typical experiment is shown. We measured about 10% of L/D positive cells in uninfected CD4TL cells.
(PPTX)

**S8 Fig. Activated effector memory CD4TL are induced to die in the presence of HIV-1-infected R5X4JT cells (Related to Fig 6). A** The percentage of R5X4JT cells infected with each of the indicated viruses was determined by flow cytometry after Gag p24 staining, two days post-inoculation. **B** PHA/IL-2-treated CD4TL from healthy donors were stained with the CellTracker (CT) dye and then cocultured with the infected R5X4JT cells at a 1/1 ratio. Coculture experiments of CD4TL with uninfected R5X4JT cells served as control. **C** After two days of coculture, CT negative (R5X4JT) and positive (CD4TL) cells were stained for expression of Gag p24, CCR7 and CD45RA, apoptosis (annexinV staining) and necrosis (Live/Dead staining). Within the fraction of gag p24 negative (bystander) CD4TL (92.9%), two cell populations

were distinguished according to their SSC-A and FSC-A parameters. Within the population of CD4TL with high granularity (SSC-high cells, 25.8% of total CD4TL), Tem cells (CCR7 negative, CD45RA negative) were selectively depleted in the presence of infected R5X4JT cells, but not in the presence of uninfected R5X4JT cells (41.7% of SSC-high cells). This effect was more marked in the presence of R5X4JT cells infected with RES viruses (5.7%), compared to SENS viruses (19.4%). The proportion of SSC-high CD4TL that are positive for cell death markers (annexin V and Live/Dead, upper panels) was also increased in the presence of RES viruses (64.2% of AnnV+ L/D+ double positive cells), compared to cells with SENS viruses (37.2%) and control cells (8.43%). Results from a typical experiment are shown. In comparison, viability of SSC-low CD4TL was only marginally influenced by infected R5X4JT cells. **D** Expression of HLA-DR was increased in SSC-high CD4TL, compared to SSC-low CD4TL. **E** Relative proportions of Tn, Tcm, Tem and Temra cells in PHA/IL-2-treated CD4TL (all) and its SSC-low and SSC-high subsets. SSC-high CD4TL were enriched in Tem cells, compared to SSC-low CD4TL.
(PPTX)

**S1 Table. Virological and immunological characteristics of individuals of the ACS harboring CXCR4-using viruses (Related to Fig 1A).**
(DOCX)

**S2 Table. Virological and immunological characteristics of individuals of the ACS harboring R5 viruses (Related to Fig 1B).**
(DOCX)

**S3 Table. CD4TL count, viral load and viral tropism in the patients who were diagnosed with HIV-1 at a chronic or advanced stage of infection (Related to Figs 1C and S1E and S1F).**
(DOCX)

**S4 Table. Baseline characteristics of patients diagnosed at the time of PHI (Related to Figs 1D and S1C, S1D, S1G and S1H).**
(DOCX)

## Acknowledgments

We greatly acknowledge Dr S. Raymond for the management of patient samples and fruitful discussions and Pr E. Bahraoui, M. Cazabat and F. Marty for technical assistance.

## Author Contributions

**Conceptualization:** Marie Armani-Tourret, Zhicheng Zhou, Jose Alcami, Fernando Arenzana-Seisdedos, Philippe Colin, Bernard Lagane.

**Formal analysis:** Marie Armani-Tourret, Zhicheng Zhou, Philippe Colin, Bernard Lagane.

**Funding acquisition:** Hugo Mouquet, Jose Alcami, Bernard Lagane.

**Investigation:** Marie Armani-Tourret, Zhicheng Zhou, Romain Gasser, Isabelle Staropoli, Vincent Cantaloube-Ferrieu, Yann Benureau, Javier Garcia-Perez, Mayte Pérez-Olmeda, Valérie Lorin, Bénédicte Puissant-Lubrano, Philippe Colin.

**Methodology:** Marie Armani-Tourret, Zhicheng Zhou, Jose Alcami, Fernando Arenzana-Seisdedos, Philippe Colin, Bernard Lagane.

**Project administration:** Bernard Lagane.

**Resources:** Lambert Assoumou, Constance Delaugerre, Jean-Daniel Lelièvre, Yves Lévy, Hugo Mouquet, Guillaume Martin-Blondel, Jose Alcami, Fernando Arenzana-Seisdedos, Jacques Izopet, Bernard Lagane.

**Supervision:** Bernard Lagane.

**Validation:** Marie Armani-Tourret, Zhicheng Zhou, Philippe Colin, Bernard Lagane.

**Visualization:** Marie Armani-Tourret, Philippe Colin, Bernard Lagane.

**Writing – original draft:** Bernard Lagane.

**Writing – review & editing:** Marie Armani-Tourret, Zhicheng Zhou, Romain Gasser, Isabelle Staropoli, Vincent Cantaloube-Ferrieu, Yann Benureau, Javier Garcia-Perez, Mayte Pérez-Olmeda, Valérie Lorin, Bénédicte Puissant-Lubrano, Lambert Assoumou, Constance Delaugerre, Jean-Daniel Lelièvre, Yves Lévy, Hugo Mouquet, Guillaume Martin-Blondel, Jose Alcami, Fernando Arenzana-Seisdedos, Jacques Izopet, Philippe Colin, Bernard Lagane.

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
