## [Decision Letter · Decision Letter 0]

27 Jan 2021

Dear Lagane,

Thank you very much for submitting your manuscript "The HIV-1 strains that evade the antiviral activity of the chemokine CXCL12 are highly pathogenic" for consideration at PLOS Pathogens. I apologize for the long review  process. The combination of seasonal holidays and the corona crisis is challenging to secure timely reviews.

As with all papers reviewed by the journal, your manuscript was reviewed by members of the editorial board and by several independent reviewers. In light of the reviews (below this email and in the attached file), we would like to invite the resubmission of a significantly-revised version that takes into account the reviewers' comments.

We cannot make any decision about publication until we have seen the revised manuscript and your response to the reviewers' comments. Your revised manuscript is also likely to be sent to reviewers for further evaluation.

Sincerely,

Alexandra Trkola

Associate Editor

PLOS Pathogens

Thomas Hope

Section Editor

PLOS Pathogens

Kasturi Haldar

Editor-in-Chief

PLOS Pathogens

orcid.org/0000-0001-5065-158X

Michael Malim

Editor-in-Chief

PLOS Pathogens

orcid.org/0000-0002-7699-2064

Reviewer's Responses to Questions

**Part I - Summary**

Reviewer #1: By way of background, HIV-1 strains that use CCR5 as a coreceptor establish the vast majority of infections in humans, and viruses that are R5 tropic tend to predominate in the early years following infection though many individuals eventually progress to AIDS with R5 viruses still predominating. However, some HIV strains evolve to use CXCR4 as a coreceptor, often while retaining the ability to use CCR5 though viruses that are only X4-tropic have certainly been described. As CXCR4 is expressed more broadly on human T cells than CCR5, this shift in coreceptor use is linked to altered cellular tropism and progression to disease. In general, the appearance of viruses that can use CXCR4 is associated with accelerated disease progression. However, this is not always the case, suggesting that there may be differences in the virulence of different X4-using viruses, though it can be difficult to unwrap viral virulence from host factors such as differences in immune responses or chemokine levels. In any event, the goal of this study is to carefully examine a panel of HIV-1 envelope proteins that use CXCR4 and that are derived from patients at different stages of infection and then to examine their properties.

There is quite a bit to digest in this study as the authors have performed a great deal of work using an impressive array of assays. A small panel of X4 envelope (Envs) proteins were cloned from patients early or late in infection (from the well-known Amsterdam cohort), virus pseudotypes constructed and different infection and protein binding assays performed. The central experimental conclusions are:

• The Envs exhibited considerable differences in sensitivity to CXCL12, the chemokine that binds to CXCR4 and so that can prevent infection via this receptor.

• Envs resistant to CXCL12 were frequent in patients with low CD4 T cell levels (late in infection), and were rare at early infection stages.

• Relative to CXCL12-sensitive Envs, resistant-Envs exhibited different antigenicity to specific neutralizing antibodies, and appeared able to utilize CXCR4 conformations that bind CXCL12 poorly thus ascribing a mechanism to the aforementioned observations.

Putting these (and other) observations together, the authors posit that X4 viruses that evolve resistance to CXCL12 have an enhanced capacity to enter naïve CD4 T cells in lymphoid organs even in the presence of this chemokine, thus making these viruses better able to dysregulate CD4T cell homeostasis and cause immunodeficiency, thus making them more pathogenic.

I think the authors are addressing an interesting topic in a way that others have not, and they have made considerable progress. I struggled a bit in assimilating this study as it seems to me that there are two very worthy goals. The first goal is to determine if CXCL12 resistant viruses are linked to accelerated disease progression, and for this you need numbers: a reasonable number of X4 viruses need to be examined from different patients to see if there is in fact a correlation. It is here that the study falls short in that only a very small number of patients and Envs were examined – the results certainly suggest that there may be a correlation, but looking at only 5 early stage and 5 late stage patients in my opinion does not justify the conclusion stated in the title of the paper, and it is this goal that makes the study appropriate for a more general audience.

The second goal is to look at the CXCL12 resistance mechanism, and in my opinion this is the strongest part of the paper. For this, you don’t need large number of Envs, but rather a representative subset that are then subjected to very detailed studies. I certainly commend the authors for the many and varied assays they employed – a number of these are not easy! I am quite happy with how the second goal is addressed, with these data being of greater interest to the real HIV experts.

A few technical points:

I think the study relies only on virus pseudotype assays for infections, which raises the possibility that differences in Env expression and virus incorporation could influence some of the results. It would be too painful to generate replication competent viruses in sufficient numbers, so are the authors reasonably confident that the results are not confounded by Env incorporation differences?

Some nice competition binding assays are performed, and one of the conclusions is that resistant gp120 proteins bind to forms of CXCR4 that sensitive gp120 proteins do not. If this is correct, then would it not follow that sensitive gp120 proteiåns could not prevent resistant gp120 proteins from binding to CXCR4?

Reviewer #2: (No Response)

**Part II – Major Issues: Key Experiments Required for Acceptance**

Reviewer #1: .

Reviewer #2: (No Response)

**Part III – Minor Issues: Editorial and Data Presentation Modifications**

Reviewer #1: .

Reviewer #2: (No Response)

PLOS authors have the option to publish the peer review history of their article (what does this mean?). If published, this will include your full peer review and any attached files.

Reviewer #1: No

Reviewer #2: No
---

## [Editor Report · Decision Letter 1]

1 Apr 2021

Dear Lagane,

We are pleased to inform you that your manuscript 'The HIV-1 strains that evade the antiviral activity of the chemokine CXCL12 are highly pathogenic' has been provisionally accepted for publication in PLOS Pathogens.

We request however that you adapt the title as suggested by the reviewers. We propose that use the following title:

Mechanisms of HIV-1 evasion to the antiviral activity of chemokine CXCL12 indicate potential links with pathogenesis

Best regards,

Alexandra Trkola

Associate Editor

PLOS Pathogens

Thomas Hope

Section Editor

PLOS Pathogens

Kasturi Haldar

Editor-in-Chief

PLOS Pathogens

orcid.org/0000-0001-5065-158X

Michael Malim

Editor-in-Chief

PLOS Pathogens

orcid.org/0000-0002-7699-2064
---

## [Editor Report · Acceptance letter]

13 Apr 2021

Dear Lagane,

We are delighted to inform you that your manuscript, "Mechanisms of HIV-1 evasion to the antiviral activity of chemokine CXCL12 indicate potential links with pathogenesis," has been formally accepted for publication in PLOS Pathogens.

Best regards,

Kasturi Haldar

Editor-in-Chief

PLOS Pathogens

orcid.org/0000-0001-5065-158X

Michael Malim

Editor-in-Chief

PLOS Pathogens

orcid.org/0000-0002-7699-2064